# A polynomial-time algorithm for learning nonparametric causal graphs

**Ming Gao**
University of Chicago
minggao@uchicago.edu

**Yi Ding**
University of Chicago
dingy@uchicago.edu

**Bryon Aragam**
University of Chicago
bryon@chicagobooth.edu

## Abstract

We establish finite-sample guarantees for a polynomial-time algorithm for learning a nonlinear, nonparametric directed acyclic graphical (DAG) model from data. The analysis is model-free and does not assume linearity, additivity, independent noise, or faithfulness. Instead, we impose a condition on the residual variances that is closely related to previous work on linear models with equal variances. Compared to an optimal algorithm with oracle knowledge of the variable ordering, the additional cost of the algorithm is linear in the dimension $d$ and the number of samples $n$. Finally, we compare the proposed algorithm to existing approaches in a simulation study.

## 1 Introduction

Modern machine learning (ML) methods are driven by complex, high-dimensional, and nonparametric models that can capture highly nonlinear phenomena. These models have proven useful in wide-ranging applications including vision, robotics, medicine, and natural language. At the same time, the complexity of these methods often obscure their decisions and in many cases can lead to wrong decisions by failing to properly account for—among other things—spurious correlations, adversarial vulnerability, and invariances [5, 7, 50]. This has led to a growing literature on correcting these problems in ML systems. A particular example of this that has received widespread attention in recent years is the problem of causal inference, which is closely related to these issues. While substantial methodological progress has been made towards embedding complex methods such as deep neural networks and RKHS embeddings into learning causal graphical models [21, 26, 31, 34, 61, 64, 65], theoretical progress has been slower and typically reserved for particular parametric models such as linear [1–3, 9, 15, 16, 29, 58, 59], generalized linear models [37, 40], and discrete models [6, 66].

In this paper, we study the problem of learning directed acyclic graphs (DAGs) from data in a nonparametric setting. Unlike existing work on this problem, we do not require linearity, additivity, independent noise, or faithfulness. Our approach is model-free and nonparametric, and uses non-parametric estimators (kernel smoothers, neural networks, splines, etc.) as "plug-in" estimators. As such, it is agnostic to the choice of nonparametric estimator chosen. Unlike existing consistency theory in the nonparametric setting [8, 20, 21, 35, 45, 49, 56], we provide explicit (nonasymptotic) finite sample complexity bounds and show that the resulting method has polynomial time complexity. The method we study is closely related to existing algorithms that first construct a variable ordering [9, 15, 16, 39]. Despite this being a well-studied problem, to the best of our knowledge our analysis is the first to provide explicit, simultaneous statistical and computational guarantees for learning nonparametric DAGs.

**Contributions** Figure 1a illustrates a key motivation for our work: While there exist methods that obtain various statistical guarantees, they lack provably efficient algorithms, or vice versa. As a result, these methods can fail in simple settings. Our focus is on *simultaneous* computational and statistical

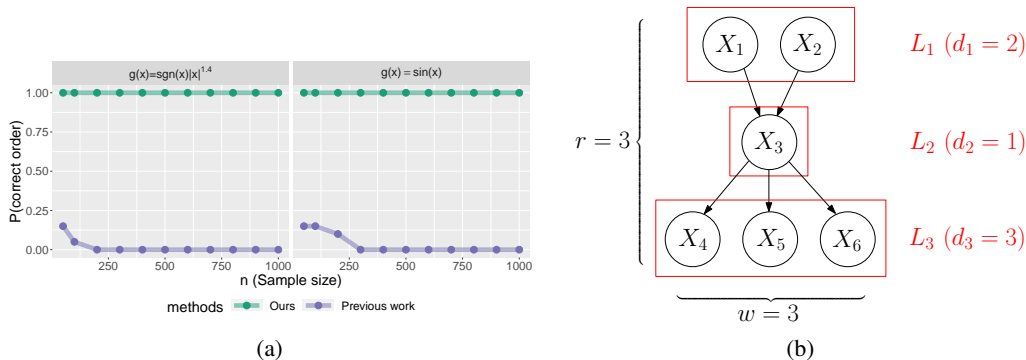

Figure 1: (a) Existing methods may not find a correct topological ordering in simple settings when $d = 3$. (b) Example of a layer decomposition $L(\mathsf{G})$ of a DAG on $d = 6$ nodes.

guarantees that are explicit and nonasymptotic in a model-free setting. More specifically, our main contributions are as follows:

- We show that the algorithms of Ghoshal and Honorio [15] and Chen et al. [9] rigorously extend to a model-free setting, and provide a method-agnostic analysis of the resulting extension (Theorem 4.1). That is, the time and sample complexity bounds depend on the choice of estimator used, and this dependence is made explicit in the bounds (Section 3.2, Section 4).

- We prove that this algorithm runs in at most $O(nd^5)$ time and needs at most $\Omega((d^2/\varepsilon)^{1+d/2})$ samples (Corollary 4.2). Moreover, the exponential dependence on $d$ can be improved by imposing additional sparsity or smoothness assumptions, and can even be made polynomial (see Section 4 for discussion). This is an expected consequence of our estimator-agnostic approach.

- We show how existing identifiability results based on ordering variances can be unified and generalized to include model-free families (Theorem 3.1, Section 3.1).

- We show that greedy algorithms such as those used in the CAM algorithm [8] can provably fail to recover an identifiable DAG (Example 5), as shown in Figure 1a (Section 3.3).

- Finally, we run a simulation study to evaluate the resulting algorithm in a variety of settings against seven state-of-the-art algorithms (Section 5).

Our simulation results can be summarized as follows: When implemented using generalized additive models [19], our method outperforms most state-of-the-art methods, particularly on denser graphs with hub nodes. We emphasize here, however, that our main contributions lay in the theoretical analysis, specifically providing a polynomial-time algorithm with sample complexity guarantees.

**Related work** The literature on learning DAGs is vast, so we focus only on related work in the nonparametric setting. The most closely related line work considers additive noise models (ANMs) [8, 10, 20, 25, 45], and prove a variety of identifiability and consistency guarantees. Compared to our work, the identifiability results proved in these papers require that the structural equations are (a) nonlinear with (b) additive, independent noise. Crucially, these papers focus on (generally asymptotic) *statistical* guarantees without any computational or algorithmic guarantees. There is also a closely related line of work for bivariate models [31–33, 60] as well as the post-nonlinear model [62]. Huang et al. [21] proposed a greedy search algorithm using an RKHS-based generalized score, and proves its consistency assuming faithfulness. Rothenhäusler et al. [49] study identifiability of a general family of partially linear models and prove consistency of a score-based search procedure in finding an equivalence class of structures. There is also a recent line of work on embedding neural networks and other nonparametric estimators into causal search algorithms [26, 34, 61, 64, 65] without theoretical guarantees. While this work was in preparation, we were made aware of the recent work [38] that proposes an algorithm that is similar to ours—also based on [15] and [9]—and establishes its sample complexity for linear Gaussian models. In comparison to these existing lines of work, our focus is on simultaneous computational and statistical guarantees that are explicit and nonasymptotic (i.e. valid for all finite $d$ and $n$), for the fully nonlinear, nonparametric, and model-free setting.

**Notation** Subscripts (e.g. $X_j$) will always be used to index random variables and superscripts (e.g. $X_j^{(i)}$) to index observations. For a matrix $W = (w_{kj})$, $w_{\cdot j} \in \mathbb{R}^d$ is the $j$th column of $W$. We denote the indices by $[d] = \{1, \ldots, d\}$, and frequently abuse notation by identifying the indices $[d]$ with the random vector $X = (X_1, \ldots, X_d)$. For example, nodes $X_j$ are interchangeable with their indices $j$ (and subsets thereof), so e.g. $\mathrm{var}(j \mid A)$ is the same as $\mathrm{var}(X_j \mid X_A)$.

## 2 Background

Let $X = (X_1, \ldots, X_d)$ be a $d$-dimensional random vector and $\mathsf{G} = (V, E)$ a DAG where we implicitly assume $V = X$. The *parent set* of a node is defined as $\mathrm{pa}_{\mathsf{G}}(X_j) = \{i : (i, j) \in E\}$, or simply $\mathrm{pa}(j)$ for short. A *source* node is any node $X_j$ such that $\mathrm{pa}(j) = \emptyset$ and an *ancestral set* is any set $A \subset V$ such that $X_j \in A \implies \mathrm{pa}(j) \subset A$. The graph $\mathsf{G}$ is called a *Bayesian network* (BN) for $X$ if it satisfies the Markov condition, i.e. that each variable is conditionally independent of its non-descendants given its parents. Intuitively, a BN for $X$ can be interpreted as a representation of the direct and indirect relationships between the $X_j$, e.g. an edge $X_i \to X_j$ indicates that $X_j$ depends directly on $X_i$, and not vice versa. Under additional assumptions such as causal minimality and no unmeasured confounding, these arrows may be interpreted causally; for more details, see the surveys [7, 50] or the textbooks [24, 28, 41, 46, 55].

The goal of structure learning is to learn a DAG $\mathsf{G}$ from i.i.d. observations $X^{(i)} \overset{\mathrm{iid}}{\sim} \mathbb{P}(X)$. Throughout this paper, we shall exploit the following well-known fact: To learn $\mathsf{G}$, it suffices to learn a topological sort of $\mathsf{G}$, i.e. an ordering $\prec$ such that $X_i \to X_j \implies X_i \prec X_j$. A brief review of this material can be found in the supplement.

**Equal variances** Recently, a new approach has emerged which was originally cast as an approach to learn equal variance DAGs [9, 15], although it has since been generalized beyond the equal variance case [16, 38, 39]. An equal variance DAG is a linear structural equation model (SEM) that satisfies

$$X_j = \langle w_{\cdot j}, X \rangle + z_j, \quad \mathrm{var}(z_j) = \sigma^2, \quad z_j \perp\!\!\!\perp \mathrm{pa}(j), \quad w_{kj} = 0 \iff k \notin \mathrm{pa}(j) \tag{1}$$

for some weights $w_{kj} \in \mathbb{R}$. Under the model (1), a simple algorithm can learn the graph $\mathsf{G}$ by first learning a topological sort $\prec$. For these models, we have the following decomposition of the variance:

$$\mathrm{var}(X_j) = \mathrm{var}(\langle w_{\cdot j}, X \rangle) + \mathrm{var}(z_j). \tag{2}$$

Thus, as long as $\mathrm{var}(\langle w_{\cdot j}, X \rangle) > 0$, we have $\mathrm{var}(X_j) > \mathrm{var}(z_j)$. It follows that as long as $\mathrm{var}(z_j)$ does not depend on $j$, it is possible to identify a source in $\mathsf{G}$ by simply minimizing the residual variances. This is the essential idea behind algorithms based on equal variances in the linear setting [9, 15]. Alternatively, it is possible to iteratively identify best sinks by minimizing marginal precisions. Moreover, this argument shows that the assumption of linearity is not crucial, and this idea can readily be extended to ANMs, as in [38]. Indeed, the crucial assumption in this argument is the independence of the noise $z_j$ and the parents $\mathrm{pa}(X_j)$; in the next section we show how these assumptions can be removed altogether.

**Layer decomposition of a DAG** Given a DAG $\mathsf{G}$, define a collection of sets as follows: $L_0 := \emptyset$, $A_j = \cup_{m=0}^j L_m$ and for $j > 0$, $L_j$ is the set of all source nodes in the subgraph $\mathsf{G}[V - A_{j-1}]$ formed by removing the nodes in $A_{j-1}$. So, e.g., $L_1$ is the set of source nodes in $\mathsf{G}$ and $A_1 = L_1$. This decomposes $\mathsf{G}$ into layers, where each layer $L_j$ consists of nodes that are sources in the subgraph $\mathsf{G}[V - A_{j-1}]$, and $A_j$ is an ancestral set for each $j$. Let $r$ denote the number of "layers" in $\mathsf{G}$, $L(\mathsf{G}) := (L_1, \ldots, L_r)$ be the corresponding layers. The quantity $r$ effectively measure the depth of a DAG. See Figure 1b for an illustration.

Learning $\mathsf{G}$ is equivalent to learning the sets $L_1, \ldots, L_r$, since any topological sort $\pi$ of $\mathsf{G}$ can be determined from $L(\mathsf{G})$, and from any sort $\pi$, the graph $\mathsf{G}$ can be recovered via variable selection. Unlike a topological sort of $\mathsf{G}$, which may not be unique, the layer decomposition $L(\mathsf{G})$ is always unique. Therefore, without loss of generality, in the sequel we consider the problem of identifying and learning $L(\mathsf{G})$.

## 3 Identifiability and algorithmic consequences

This section sets the stage for our main results on learning nonparametric DAGs: First, we show that existing identifiability results for equal variances generalize to a family of model-free, nonparametric distributions. Second, we show that this motivates an algorithm very similar to existing algorithms in the equal variance case. We emphasize that various incarnations of these ideas have appeared in previous work [9, 15, 16, 38, 39], and our effort in this section is to unify these ideas and show that the same ideas can be applied in more general settings without linearity or independent noise. Once this has been done, our main sample complexity result is presented in Section 4.

### 3.1 Nonparametric identifiability

In general, a BN for $X$ need not be unique, i.e. $\mathsf{G}$ is not necessarily identifiable from $\mathbb{P}(X)$. A common strategy in the literature to enforce identifiability is to impose structural assumptions on the conditional distributions $\mathbb{P}(X_j \mid \mathrm{pa}(j))$, for which there is a broad literature on identifiability. Our first result shows that identifiability is guaranteed as long as the residual variances $\mathbb{E}\,\mathrm{var}(X_j \mid \mathrm{pa}(j))$ do not depend on $j$. This is a natural generalization of the notion of equality of variances for linear models [9, 15, 44].

**Theorem 3.1.** *If $\mathbb{E}\,\mathrm{var}(X_j \mid \mathrm{pa}(j)) \equiv \sigma^2$ does not depend on $j$, then $\mathsf{G}$ is identifiable from $\mathbb{P}(X)$.*

The proof of Theorem 3.1 can be found in the supplement. This result makes no structural assumptions on the local conditional probabilities $\mathbb{P}(X_j \mid \mathrm{pa}(j))$. To illustrate, we consider some examples below.

**Example 1** (Causal pairs, [33])**.** Consider a simple model on two variables: $X \to Y$ with $\mathbb{E}\,\mathrm{var}(Y \mid X) = \mathrm{var}(X)$. Then as long as $\mathbb{E}[Y \mid X]$ is nonconstant, Theorem 3.1 implies the causal order is identifiable. No additional assumptions on the noise or functional relationships are necessary.

**Example 2** (Binomial models, [40])**.** Assume $X_j \in \{0, 1\}$ and $X_j = \mathrm{Ber}(f_j(\mathrm{pa}(j)))$ with $f_j(\mathrm{pa}(j)) \in [0, 1]$. Then Theorem 3.1 implies that if $\mathbb{E}f_j(\mathrm{pa}(j))(1 - f_j(\mathrm{pa}(j))) \equiv \sigma^2$ does not depend on $j$, then $\mathsf{G}$ is identifiable.

**Example 3** (Generalized linear models)**.** The previous example can of course be generalized to arbitrary generalized linear models: Assume $\mathbb{P}[X_j \mid \mathrm{pa}(j)] \propto \exp(X_j\theta_j - K(\theta_j))$, where $\theta_j = f_j(\mathrm{pa}(j))$ and $K(\theta_j)$ is the partition function. Then Theorem 3.1 implies that if $\mathbb{E}[K''(f_j(\mathrm{pa}(j)))] \equiv \sigma^2$ does not depend on $j$, then $\mathsf{G}$ is identifiable.

**Example 4** (Additive noise models, [45])**.** Finally, we observe that Theorem 3.1 generalizes existing results for ANMs: In an ANM, we have $X_j = f_j(\mathrm{pa}(j)) + z_j$ with $z_j \perp\!\!\!\perp \mathrm{pa}(j)$. If $\mathrm{var}(z_j) = \sigma^2$, then an argument similar to (2) shows that ANMs with equal variances are identifiable. Theorem 3.1 applies to more general additive noise models $X_j = f_j(\mathrm{pa}(j)) + g_j(\mathrm{pa}(j))^{1/2}z_j$ with heteroskedastic, uncorrelated (i.e. not necessarily independent) noise.

**Unequal variances** Early work on this problem focused on the case of equal variances [9, 15], as we have done here. This assumption illustrates the main technical difficulties in proving identifiability, and it is well-known by now that equality of variances is not necessary, and a weaker assumption that allows for heterogeneous residual variances suffices in special cases [16, 39]. Similarly, the extension of Theorem 3.1 to such heterogeneous models is straightforward, and omitted for brevity; see Appendix B.1 in the supplement for additional discussion and simulations. In the sequel, we focus on the case of equality for simplicity and ease of interpretation.

### 3.2 A polynomial-time algorithm

The basic idea behind the top-down algorithm proposed in [9] can easily be extended to the setting of Theorem 3.1, and is outlined in Algorithm 1. The only modification is to replace the error variances $\mathrm{var}(z_j) = \sigma^2$ from the linear model (1) with the corresponding residual variances (i.e. $\mathbb{E}\,\mathrm{var}(X_\ell \mid S_j)$), which are well-defined for any $\mathbb{P}(X)$ with finite second moments.

A natural idea to translate Algorithm 1 into an empirical algorithm is to replace the residual variances with an estimate based on the data. One might then hope to use similar arguments as in the linear setting to establish consistency and bound the sample complexity. Perhaps surprisingly, this does not work unless the topological sort of $\mathsf{G}$ is unique. When there is more than one topological sort, it becomes necessary to uniformly bound the errors of all possible residual variances—and in the worst

---

**Algorithm 1** Population algorithm for learning nonparametric DAGs

---

1. Set $S_0 = \emptyset$ and for $j = 0, 1, 2, \ldots$, let

$$k_j = \underset{\ell \notin S_j}{\arg\min} \, \mathbb{E} \, \mathrm{var}(X_\ell \,|\, S_j), \qquad S_{j+1} = S_j \cup \{k_j\}.$$

2. Return the DAG $\mathsf{G}$ that corresponds to the topological sort $(k_1, \ldots, k_d)$.

---

---

**Algorithm 2** NPVAR algorithm

---

**Input:** $X^{(1)}, \ldots, X^{(n)}, \eta > 0$.

1. Set $\widehat{L}_0 = \emptyset$, $\widehat{\sigma}_{\ell 0}^2 = \widehat{\mathrm{var}}(X_\ell)$, $k_0 = \arg\min_\ell \widehat{\sigma}_{\ell 0}^2$, $\widehat{\sigma}_0^2 = \sigma_{k_0 0}^2$.

2. Set $\widehat{L}_1 := \{\ell : |\widehat{\sigma}_{\ell 0}^2 - \widehat{\sigma}_0^2| < \eta\}$.

3. For $j = 2, 3, \ldots$:

   (a) Randomly split the $n$ samples in half and let $\widehat{A}_j := \cup_{m=1}^j \widehat{L}_m$.

   (b) For each $\ell \notin \widehat{A}_j$, use the first half of the sample to estimate $f_{\ell j}(X_{\widehat{A}_j}) = \mathbb{E}[X_\ell \,|\, \widehat{A}_j]$ via a nonparametric estimator $\widehat{f}_{\ell j}$.

   (c) For each $\ell \notin \widehat{A}_j$, use the second half of the sample to estimate the residual variances via the plug-in estimator

   $$\widehat{\sigma}_{\ell j}^2 = \frac{1}{n/2} \sum_{i=1}^{n/2} (X_\ell^{(i)})^2 - \frac{1}{n/2} \sum_{i=1}^{n/2} \widehat{f}_{\ell j}(X_{\widehat{A}_j}^{(i)})^2. \tag{3}$$

   (d) Set $k_j = \arg\min_{\ell \notin \widehat{A}_j} \widehat{\sigma}_{\ell j}^2$ and $\widehat{L}_{j+1} = \{\ell : |\widehat{\sigma}_{\ell j}^2 - \widehat{\sigma}_{k_j j}^2| < \eta, \ell \notin \widehat{A}_j\}$.

4. Return $\widehat{L} = (\widehat{L}_1, \ldots, \widehat{L}_{\widehat{r}})$.

---

case there are exponentially many ($d2^{d-1}$ to be precise) possible residual variances. The key issue is that the sets $S_j$ in Algorithm 1 are *random* (i.e. data-dependent), and hence unknown in advance. This highlights a key difference between our algorithm and existing work for linear models such as [9, 15, 16, 38]: In our setting, the residual variances cannot be written as simple functions of the covariance matrix $\Sigma := \mathbb{E}XX^T$, which simplifies the analysis for linear models considerably. Indeed, although the same exponential blowup arises for linear models, in that case consistent estimation of the covariance matrix $\Sigma := \mathbb{E}XX^T$ provides *uniform* control over all possible residual variances (e.g., see Lemma 6 in [9]). In the nonparametric setting, this reduction no longer applies.

To get around this technical issue, we modify Algorithm 1 to learn $\mathsf{G}$ one layer $L_j$ at a time, as outlined in Algorithm 2 (see Section 2 for details on $L_j$). As a result, we need only estimate $\sigma_{\ell j}^2 := \mathbb{E} \, \mathrm{var}(X_\ell \,|\, A_j)$, which involves regression problems with at most $|A_j|$ nodes. We use the plug-in estimator (3) for this, although more sophisticated estimators are available [14, 47]. This also necessitates the use of sample splitting in Step 3(a) of Algorithm 2, which is necessary for the theoretical arguments but not needed in practice.

The overall computational complexity of Algorithm 2, which we call NPVAR, is $O(ndrT)$, where $T$ is the complexity of computing each nonparametric regression function $\widehat{f}_{\ell j}$. For example, if a kernel smoother is used, $T = O(d^3)$ and thus the overall complexity is $O(nrd^4)$. For comparison, an oracle algorithm that knows the true topological order of $\mathsf{G}$ in advance would still need to compute $d$ regression functions, and hence would have complexity $O(dT)$. Thus, the extra complexity of learning the topological order is only $O(nr) = O(nd)$, which is linear in the dimension and the number of samples. Furthermore, under additional assumptions on the sparsity and/or structure of the DAG, the time complexity can be reduced further, however, our analysis makes no such assumptions.

### 3.3 Comparison to existing algorithms

Compared to existing algorithms based on order search and equal variances, NPVAR applies to more general models without parametric assumptions, independent noise, or additivity. It is also instructive to make comparisons with greedy score-based algorithms such as causal additive models (CAM, [8]) and greedy DAG search (GDS, [44]). We focus here on CAM since it is more recent and applies in nonparametric settings, however, similar claims apply to GDS as well.

CAM is based around greedily minimizing the log-likelihood score for additive models with Gaussian noise. In particular, it is not guaranteed to find a global minimizer, which is as expected since it is based on a nonconvex program. This is despite the global minimizer—if it can be found—having good statistical properties. The next example shows that, in fact, there are identifiable models for which CAM will find the wrong graph with high probability.

**Example 5.** Consider the following three-node additive noise model with $z_j \sim \mathcal{N}(0, 1)$:

$$\begin{aligned} X_1 &= z_1, \\ X_2 &= g(X_1) + z_2, \\ X_3 &= g(X_1) + g(X_2) + z_3. \end{aligned} \tag{4}$$

In the supplement (Appendix D), we show the following: *There exist infinitely many nonlinear functions $g$ for which the CAM algorithm returns an incorrect order under the model* (4). This is illustrated empirically in Figure 1a for the nonlinearities $g(u) = \mathrm{sgn}(u)|u|^{1.4}$ and $g(u) = \sin u$. In each of these examples, the model satisfies the identifiability conditions for CAM as well as the conditions required in our work.

We stress that this example does not contradict the statistical results in Bühlmann et al. [8]: It only shows that the *algorithm* may not find a global minimizer and as a result, returns an incorrect variable ordering. Correcting this discrepancy between the algorithmic and statistical results is a key motivation behind our work. In the next section, we show that NPVAR provably learns the true ordering—and hence the true DAG—with high probability.

## 4 Sample complexity

Our main result analyzes the sample complexity of NPVAR (Algorithm 2). Recall the layer decomposition $L(\mathsf{G})$ from Section 2 and define $d_j := |A_j|$. Let $f_{\ell j}(X_{A_j}) = \mathbb{E}[X_\ell \,|\, A_j]$.

**Condition 1** (Regularity). For all $j$ and all $\ell \notin A_j$, (a) $X_j \in [0, 1]$, (b) $f_{\ell j} : [0, 1]^{d_j} \to [0, 1]$, (c) $f_{\ell j} \in L^\infty([0, 1]^{d_j})$, and (d) $\mathrm{var}(X_\ell \,|\, A_j) \le \zeta_0 < \infty$.

These are the standard regularity conditions from the literature on nonparametric statistics [18, 57], and can be weakened (e.g. if the $X_j$ and $f_{\ell j}$ are unbounded, see [23]). We impose these stronger assumptions in order to simplify the statements and focus on technical details pertinent to graphical modeling and structure learning. The next assumption is justified by Theorem 3.1, and as we have noted, can also be weakened.

**Condition 2** (Identifiability). $\mathbb{E}\,\mathrm{var}(X_j \,|\, \mathrm{pa}(j)) \equiv \sigma^2$ does not depend on $j$.

Our final condition imposes some basic finiteness and consistency requirements on the chosen nonparametric estimator $\widehat{f}$, which we view as a function for estimating $\mathbb{E}[Y \,|\, Z]$ from an arbitrary distribution over the pair $(Y, Z)$.

**Condition 3** (Estimator). The nonparametric estimator $\widehat{f}$ satisfies (a) $\mathbb{E}[Y \,|\, Z] \in L^\infty \implies \widehat{f} \in L^\infty$ and (b) $\mathbb{E}_{\widehat{f}} \|\widehat{f}(Z) - \mathbb{E}[Y \,|\, Z]\|_2^2 \to 0$.

This is a mild condition that is satisfied by most popular estimators including kernel smoothers, nearest neighbours, and splines, and in particular, Condition 3(a) is only used to simplify the theorem statement and can easily be relaxed.

**Theorem 4.1.** *Assume Conditions 1-3. Let $\Delta_j > 0$ be such that $\mathbb{E}\,\mathrm{var}(X_\ell \,|\, A_j) > \sigma^2 + \Delta_j$ for all $\ell \notin A_j$ and define $\Delta := \inf_j \Delta_j$. Let $\delta^2 := \sup_{\ell, j} \mathbb{E}_{\widehat{f}_{\ell j}} \|f_{\ell j}(X_{A_j}) - \widehat{f}_{\ell j}(X_{A_j})\|_2^2$. Then for any*

$\delta\sqrt{d} < \eta < \Delta/2$,

$$\mathbb{P}(\widehat{L} = L(\mathsf{G})) \gtrsim 1 - \frac{\delta^2}{\eta^2}rd \tag{5}$$

Once the layer decomposition $L(\mathsf{G})$ is known, the graph $\mathsf{G}$ can be learned via standard nonlinear variable selection methods (see Appendix A in the supplement).

A feature of this result is that it is agnostic to the choice of estimator $\widehat{f}$, as long as it satisfies Condition 3. The dependence on $\widehat{f}$ is quantified through $\delta^2$, which depends on the sample size $n$ and represents the rate of convergence of the chosen nonparametric estimator. Instead of choosing a specific estimator, Theorem 4.1 is stated so that it can be applied to general estimators. As an example, suppose each $f_{\ell j}$ is Lipschitz continuous and $\widehat{f}$ is a standard kernel smoother. Then

$$\mathbb{E}_{\widehat{f}_{\ell j}}\|f_{\ell j}(X_{L_j}) - \widehat{f}_{\ell j}(X_{L_j})\|_2^2 \le \delta^2 \lesssim n^{-\frac{2}{2+d}}.$$

Thus we have the following special case:

**Corollary 4.2.** *Assume each $f_{\ell j}$ is Lipschitz continuous. Then $\widehat{L}$ can be computed in $O(nd^5)$ time and $\mathbb{P}(\widehat{L} = L(\mathsf{G})) \ge 1 - \varepsilon$ as long as $n = \Omega((rd/(\eta^2\varepsilon))^{1+d/2})$.*

This is the best possible rate attainable by any algorithm without imposing stronger regularity conditions (see e.g. §5 in [18]). Furthermore, $\delta^2$ can be replaced with the error of an arbitrary estimator of the residual variance itself (i.e. something besides the plug-in estimator (3)); see Proposition C.4 in Appendix C for details.

To illustrate these results, consider the problem of finding the direction of a Markov chain $X_1 \to X_2 \to \cdots \to X_d$ whose transition functions $\mathbb{E}[X_j \,|\, X_{j-1}]$ are each Lipschitz continuous. Then $r = d$, so Corollary 4.2 implies that $n = \Omega((d^2/(\eta\sqrt{\varepsilon}))^{1+d/2})$ samples are sufficient to learn the order—and hence the graph as well as each transition function—with high probability. Since $r = d$ for any Markov chain, this particular example maximizes the dependence on $d$; at the opposite extreme a bipartite graph with $r = 2$ would require only $n = \Omega((\sqrt{d}/(\eta\sqrt{\varepsilon}))^{1+d/2})$. In these lower bounds, it is not necessary to know the type of graph (e.g. Markov chain, bipartite) or the depth $r$.

**Choice of $\eta$** The lower bound $\eta > \delta\sqrt{d}$ is not strictly necessary, and is only used to simplify the lower bound in (5). In general, taking $\eta$ sufficiently small works well in practice. The main tradeoff in choosing $\eta > 0$ is computational: A smaller $\eta$ may lead to "splitting" one of the layers $L_j$. In this case, NPVAR still recovers the structure correctly, but the splitting results in redundant estimation steps in Step 3 (i.e. instead of estimating $L_j$ in one iteration, it takes multiple iterations to estimate correctly). The upper bound, however, is important: If $\eta$ is too large, then we may include spurious nodes in the layer $L_j$, which would cause problems in subsequent iterations.

**Nonparametric rates** Theorem 4.1 and Corollary 4.2 make no assumptions on the sparsity of $\mathsf{G}$ or smoothness of the mean functions $\mathbb{E}[X_\ell \,|\, A_j]$. For this reason, the best possible rate for a naïve plug-in estimator of $\mathbb{E}\,\mathrm{var}(X_\ell \,|\, A_j)$ is bounded by the minimax rate for estimating $\mathbb{E}[X_\ell \,|\, A_j]$. For practical reasons, we have chosen to focus on an agnostic analysis that does not rely on any particular estimator. Under additional sparsity and smoothness assumptions, these rates can be improved, which we briefly discuss here.

For example, by using adaptive estimators such as RODEO [27] or GRID [17], the sample complexity will depend only on the sparsity of $f_{\ell j}(X_{A_j})$, i.e. $d^* = \max_j \max_{\ell \notin A_j} |\{k \in A_j : \partial_k f_{\ell j} \ne 0\}|$, where $\partial_k$ is the $k$th partial derivative. Another approach that does not require adaptive estimation is to assume $|L_j| \le w$ and define $r^* := \sup\{|i - j| : e = (e_1, e_2) \in E, e_1 \in L_i, e_2 \in L_j\}$. Then $\delta^2 \asymp n^{-2/(2+wr^*)}$, and the resulting sample complexity depends on $wr^*$ instead of $d$. For a Markov chain with $w = r^* = 1$ this leads to a substantial improvement.

Instead of sparsity, we could impose stronger smoothness assumptions: Let $\beta_*$ denote the smallest Hölder exponent of any $f_{\ell j}$. Then if $\beta_* \ge d/2$, then one can use a one-step correction to the plug-in estimator (3) to obtain a root-$n$ consistent estimator of $\mathbb{E}\,\mathrm{var}(X_\ell \,|\, A_j)$ [22, 47]. Another approach is to use undersmoothing [14]. In this case, the exponential sample complexity improves to polynomial sample complexity. For example, in Corollary 4.2, if we replace Lipschitz with the stronger condition that $\beta_* \ge d/2$, then the sample complexity improves to $n = \Omega(rd/(\eta^2\varepsilon))$.

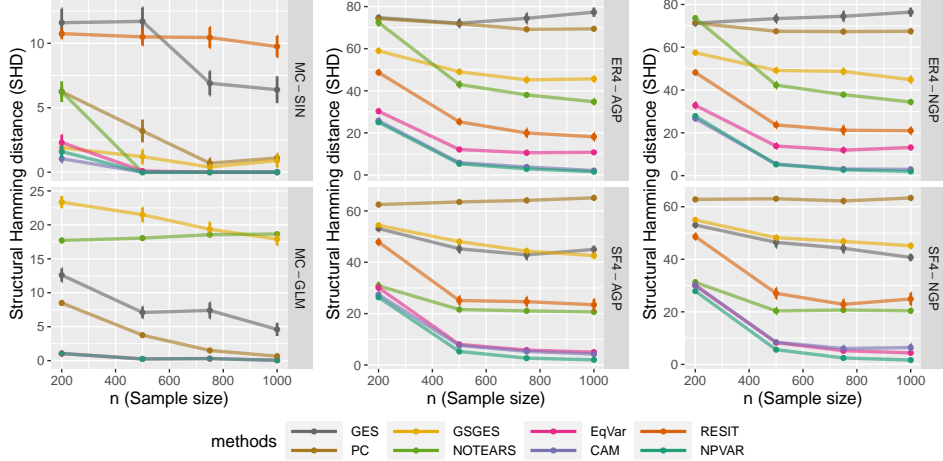

(a) SHD vs. $n$ ($d = 20$).

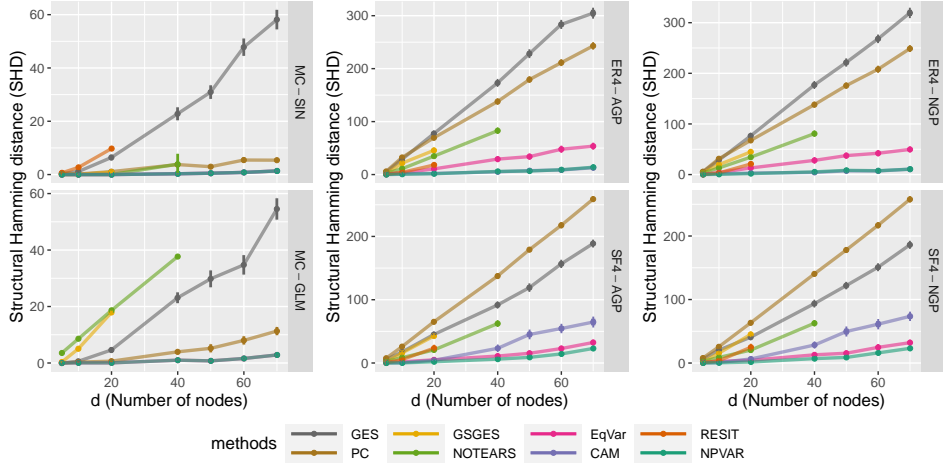

(b) SHD vs. $d$ ($n = 1000$).

Figure 2: Structural Hamming distance (SHD) as a function of sample size ($n$) and number of nodes ($d$). Error bars denote $\pm 1$ standard error. Some algorithms were only run for sufficiently small graphs due to high computational cost.

## 5 Experiments

Finally, we perform a simulation study to compare the performance of NPVAR against state-of-the-art methods for learning nonparametric DAGs. The algorithms are: RESIT [45], CAM [8], EqVar [9], NOTEARS [64], GSGES [21], PC [54], and GES [11]. In our implementation of NPVAR, we use generalized additive models (GAMs) for both estimating $\widehat{f}_{\ell j}$ and variable selection. One notable detail is our implementation of EqVar, which we adapted to the nonlinear setting by using GAMs instead of subset selection for variable selection (the order estimation step remains the same). Full details of the implementations used as well as additional experiments can be found in the supplement. Code implementing the NPVAR algorithm is publicly available at https://github.com/MingGao97/NPVAR.

We conducted a series of simulation on different graphs and models, comparing the performance in both order recovery and structure learning. Due to space limitations, only the results for structure learning in the three most difficult settings are highlighted in Figure 2. These experiments correspond to non-sparse graphs with non-additive dependence given by either a Gaussian process (GP) or a generalized linear model (GLM):

- *Graph types.* We sampled three families of DAGs: Markov chains (MC), Erdös-Rényi graphs (ER), and scale-free graphs (SF). For MC graphs, there are exactly $d$ edges, whereas for ER and SF graphs, we sample graphs with $kd$ edges on average. This is denoted by ER4/SF4 for $k = 4$ in Figure 2. Experiments on sparser DAGs can be found in the supplement.

- *Probability models.* For the Markov chain models, we used two types of transition functions: An additive sine model with $\mathbb{P}(X_j \,|\, X_{j-1}) = \mathcal{N}(\sin(X_{j-1}), \sigma^2)$ and a discrete model (GLM) with $X_j \in \{0, 1\}$ and $\mathbb{P}(X_j \,|\, X_{j-1}) \in \{p, 1 - p\}$. For the ER and SF graphs, we sampled $\mathbb{E}[X_j \,|\, \mathrm{pa}(j)]$ from both additive GPs (AGP) and non-additive GPs (NGP).

Full details as well as additional experiments on order recovery, additive models, sparse graphs, and misspecified models can be found in the supplement (Appendix E).

**Structure learning**    To evaluate overall performance, we computed the structural Hamming distance (SHD) between the learned DAG and the true DAG. SHD is a standard metric used for comparison of graphical models. According to this metric, the clear leaders are NPVAR, EqVar, and CAM. Consistent with existing results, existing methods tend to suffer as the edge density and dimension of the graphs increase, however, NPVAR is more robust in these settings. Surprisingly, the CAM algorithm remains quite competitive for non-additive models, although both EqVar and NPVAR clearly outperform CAM. On the GLM model, which illustrates a non-additive model with non-additive noise, EqVar and NPVAR performed the best, although PC showed good performance with $n = 1000$ samples. Both CAM and RESIT terminated with numerical issues on the GLM model.

These experiments serve to corroborate our theoretical results and highlight the effectiveness of the NPVAR algorithm, but of course there are tradeoffs. For example, algorithms such as CAM which exploit sparse and additive structure perform very well in settings where sparsity and additivity can be exploited, and indeed outperform NPVAR in some cases. Hopefully, these experiments can help to shed some light on when various algorithms are more or less effective.

**Misspecification and sensitivity analysis**    We also considered two cases of misspecification: In Appendix B.1, we consider an example where Condition 2 fails, but NPVAR still successfully recovers the true ordering. This experiment corroborates our claims that this condition can be relaxed to handle unequal residual variances. We also evaluated the performance of NPVAR on linear models as in (1), and in all cases it was able to recover the correct ordering.

## 6  Discussion

In this paper, we analyzed the sample complexity of a polynomial-time algorithm for estimating nonparametric causal models represented by a DAG. Notably, our analysis avoids many of the common assumptions made in the literature. Instead, we assume that the residual variances are equal, similar to assuming homoskedastic noise in a standard nonparametric regression model. Our experiments confirm that the algorithm, called NPVAR, is effective at learning identifiable causal models and outperforms many existing methods, including several recent state-of-the-art methods. Nonetheless, existing algorithms such as CAM are quite competitive and apply in settings where NPVAR does not.

We conclude by discussing some limitations and directions for future work. Although we have relaxed many of the common assumptions made in the literature, these assumptions have been replaced by an assumption on the residual variances that may not hold in practice. An interesting question is whether or not there exist provably polynomial-time algorithms for nonparametric models in under less restrictive assumptions. Furthermore, although the proposed algorithm is polynomial-time, the worst-case $O(d^5)$ dependence on the dimension is of course limiting. This can likely be reduced by developing more efficient estimators of the residual variance that do not first estimate the mean function. This idea is common in the statistics literature, however, we are not aware of such estimators specifically for the residual variance (or other nonlinear functionals of $\mathbb{P}(X)$). Furthermore, our general approach can be fruitfully applied to study various parametric models that go beyond linear models, for which both computation and sample efficiency would be expected to improve. These are interesting directions for future work.

## Broader Impact

Causality and interpretability are crucial aspects of modern machine learning systems. Graphical models in particular are a promising tool at the intersection of causality and interpretability, and our work provides an intuitive approach to balance these issues against modeling flexibility with nonparametric models. That being said, as this work is primarily theoretical, the broader impacts and ethical implications of our work are most likely to be felt downstream in applications. For example, while DAGs can provide causal insights under certain assumptions, these models can potentially be used to provide a false sense of security when they are not applied and deployed carefully. Along these lines, our work attempts to provide a rigorous sense of when flexible nonparametric causal models can be learned from data, by developing both theory and algorithms to justify these models from both mathematical and empirical perspectives.

## Acknowledgments and Disclosure of Funding

We thank the anonymous reviewers for valuable feedback, as well as Y. S. Wang and E. H. Kennedy for helpful discussions. B.A. acknowledges the support of the NSF via IIS-1956330. Y.D.'s work has been partially supported by the NSF (CCF-1439156, CCF-1823032, CNS-1764039).

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
