[Supplementary Material]

**Supplementary Material for: A polynomial-time algorithm for learning non-parametric causal graphs**

## A   Reduction to order search

The fact that DAG learning can be reduced to learning a topological sort is well-known. For example, this fact is the basis of exact algorithms for score-based learning based on dynamic programming [34, 40, 50, 51] as well as recent algorithms for linear models [8, 14, 15]. See also [49]. This fact has also been exploited in the overdispersion scoring model developed by Park and Raskutti [38] as well as for nonlinear additive models [7]. In fact, more can be said: Any ordering defines a minimal I-map of $\mathbb{P}(X)$ via a simple iterative algorithm (see §3.4.1, Algorithm 3.2 in [23]), and this minimal I-map is unique as long as $\mathbb{P}(X)$ satisfies the intersection property. This is guaranteed, for example, if $\mathbb{P}(X)$ has a positive density, but holds under weaker conditions (see [41] for necessary and sufficient conditions assuming $\mathbb{P}(X)$ has a density and [12], Theorem 7.1, for the general case). This same algorithm can then be used to reconstruct the true DAG $\mathsf{G}$ from the true ordering $\prec$. As noted in Section 2, a further reduction can be obtained by considering the layer decomposition $L(\mathsf{G})$, from which all topological orders $\prec$ of $\mathsf{G}$ can be deduced.

Once the ordering is known, existing nonlinear variable selection methods [4, 11, 16, 25, 28, 46] suffice to learn the parent sets $\mathrm{pa}(j)$ and hence the graph $\mathsf{G}$. More specifically, given an order $\prec$, to identify $\mathrm{pa}(j)$, let $f(S_j) := \mathbb{E}[X_j \,|\, S_j]$, where $S_j := \{X_k : X_k \prec X_j\}$. The parent set of $X_j$ is given by the active variables in this conditional expectation, i.e. $\mathrm{pa}(j) = \{k : \partial_k f \neq 0\}$, where $\partial_k$ is the partial derivative of $f$ with respect to the $k$th argument.

In our experiments, we use exactly this procedure to learn $\mathsf{G}$ from the order $\prec$, based on the data. Specifically, we use generalized additive models, similar to the pruning step in [7]. See Appendix E for more details.

## B   Proof of Theorem 3.1

The key lemma is the following, which is easy to prove for additive noise models via (2), and which we show holds more generally in non-additive models:

**Lemma B.1.** *Let $A \subset V$ be an ancestral set in $\mathsf{G}$. If $\mathbb{E}\,\mathrm{var}(X_j \,|\, \mathrm{pa}(j)) \equiv \sigma^2$ does not depend on $j$, then for any $j \notin A$,*

$$\mathbb{E}\,\mathrm{var}(X_j \,|\, X_A) = \sigma^2 \quad \text{if } \mathrm{pa}(j) \subset A,$$
$$\mathbb{E}\,\mathrm{var}(X_j \,|\, X_A) > \sigma^2 \quad \text{otherwise.}$$

*Proof of Lemma B.1.* Let $B_j = \mathrm{pa}(X_j)$, $\overline{B_j} := B_j - A$, and $\overline{\mathsf{G}}$ be the subgraph of $\mathsf{G}$ formed by removing the nodes in the ancestral set $A$. Then

$$\mathrm{var}(X_j \,|\, X_A) = \mathbb{E}[\mathrm{var}(X_j \,|\, X_A, X_{B_j}) \,|\, X_A] + \mathrm{var}[\mathbb{E}[X_j \,|\, X_A, X_{B_j}] \,|\, X_A]$$
$$= \mathbb{E}[\mathrm{var}(X_j \,|\, X_A, X_{\overline{B_j}}) \,|\, X_A] + \mathrm{var}[\mathbb{E}[X_j \,|\, X_A, X_{\overline{B_j}}] \,|\, X_A].$$

There are two cases: (i) $\overline{B_j} = \emptyset$, and (ii) $\overline{B_j} \neq \emptyset$. In case (i), it follows that $B_j \subset A$ and hence $\mathrm{pa}(j) \subset A$. Since $X_j$ is conditionally independent of its nondescendants (e.g. ancestors) given its parents, it follows that $\mathrm{var}(X_j \,|\, X_A) = \mathrm{var}(X_j \,|\, X_{B_j})$ and hence

$$\mathbb{E}\,\mathrm{var}(X_j \,|\, X_A) = \mathbb{E}\,\mathrm{var}(X_j \,|\, X_{B_j}) = \sigma^2.$$

In case (ii), it follows that

$$\mathrm{var}(X_j \,|\, X_A) = \mathbb{E}[\mathrm{var}(X_j \,|\, X_A, X_{\overline{B_j}}) \,|\, X_A] + \mathrm{var}[\mathbb{E}[X_j \,|\, X_A, X_{\overline{B_j}}] \,|\, X_A]$$
$$= \mathbb{E}[\mathrm{var}(X_j \,|\, X_{B_j}) \,|\, X_A] + \mathrm{var}[\mathbb{E}[X_j \,|\, X_{B_j}] \,|\, X_A],$$

where again we used that $X_j$ is conditionally independent of its nondescendants (e.g. ancestors) given its parents to replace conditioning on $(X_A, X_{\overline{B_j}}) = X_{A \cup B_j}$ with conditioning on $B_j$.

Now suppose $X_k$ is in case (i) and $X_j$ is in case (ii). We wish to show that $\mathbb{E}\operatorname{var}(X_j \mid X_A) > \mathbb{E}\operatorname{var}(X_k \mid X_A) = \sigma^2$. Then

$$
\begin{aligned}
\mathbb{E}\operatorname{var}(X_j \mid X_A) &= \mathbb{E}\big[\mathbb{E}[\operatorname{var}(X_j \mid X_{B_j}) \mid X_A]\big] + \mathbb{E}\operatorname{var}[\mathbb{E}[X_j \mid X_{B_j}] \mid X_A] \\
&> \mathbb{E}\big[\mathbb{E}[\operatorname{var}(X_j \mid \operatorname{pa}(j)) \mid X_A]\big] \\
&= \mathbb{E}\operatorname{var}(X_j \mid \operatorname{pa}(j)) \\
&= \mathbb{E}\operatorname{var}(X_k \mid \operatorname{pa}(k)) \\
&= \sigma^2,
\end{aligned}
$$

where we have invoked the assumption that $\mathbb{E}\operatorname{var}(X_j \mid \operatorname{pa}(j))$ does not depend on $j$ to conclude $\mathbb{E}[\operatorname{var}(X_j \mid \operatorname{pa}(j)) \mid X_A] = \mathbb{E}[\operatorname{var}(X_k \mid \operatorname{pa}(k)) \mid X_A]$. This completes the proof. $\qquad\square$

Theorem 3.1 is an immediate corollary of Lemma B.1. For completeness, we include a proof below.

*Proof of Theorem 3.1.* Let $S(\mathsf{G})$ denote the set of sources in $\mathsf{G}$ and note that Lemma B.1 implies that if $X_s \in S(\mathsf{G})$, then $\operatorname{var}(X_s) < \operatorname{var}(X_j)$ for any $s \neq j$. Thus, $S(\mathsf{G})$ is identifiable. Let $\mathsf{G}_1$ denote the subgraph of $\mathsf{G}$ formed by removing the nodes in $S(\mathsf{G})$. Since $S(\mathsf{G}) = L_1 = A_1$, $S(\mathsf{G})$ is an ancestral set in $\mathsf{G}$. After conditioning on $A_1$, we can thus apply Lemma B.1 once again to identify the sources in $\mathsf{G}_1$, i.e. $S(\mathsf{G}_1) = L_2$. By repeating this procedure, we can recursively identify $L_1, \ldots, L_r$, and hence any topological sort of $\mathsf{G}$. $\qquad\square$

## B.1 Generalization to unequal variances

In this appendix, we illustrate how Theorem 3.1 can be extended to the case where residual variances are different, i.e. $\sigma_j^2 = \mathbb{E}\operatorname{var}(X_j \mid \operatorname{pa}(j))$ is not independent of $j$. Let $\operatorname{de}(i)$ be the descendant of node $i$ and $[a:b] = \{a, a+1, \ldots, b-1, b\}$. Note also that for any nodes $X_u$ and $X_v$ in the same layer $L_m$ of the graph, if we interchange the position of $u$ and $v$ in some true ordering $\pi$ consistent with the graph to get $\pi_u$ and $\pi_v$, both $\pi_u$ and $\pi_v$ are correct orderings.

The following result is similar to existing results on unequal variances [15, 36, 37], with the exception that it applies to general DAGs without linearity, additivity, or independent noise.

**Theorem B.2.** *Suppose there exists an ordering $\pi$ such that for all $j \in [1:d]$ and $k \in \pi_{[j+1:d]}$, the following conditions holds:*

   *1. If $i = \pi_j$ and $k$ are not in the same layer $L_m$, then*

$$
\sigma_i^2 < \sigma_k^2 + \mathbb{E}\operatorname{var}(\mathbb{E}(X_k \mid \operatorname{pa}(k))|X_{\pi_{[1:j-1]}}). \tag{6}
$$

   *2. If $i$ and $k$ are in the same layer $L_m$, then either $\sigma_i^2 = \sigma_k^2$ or (6) holds.*

*Then the order $\pi$ is identifiable.*

In this condition not only do we need to control the descendants of a node, but also the other non-descendants that have not been identified.

Before proving this result, we illustrate it with an example.

**Example 6.** Let's consider a very simple case: A Markov chain with three nodes $X_1 \to X_2 \to X_3$ such that

$$
\begin{aligned}
X_1 &= z_1 \sim N(0, 1) \\
X_2 &= \frac{1}{2}X_1^2 + z_2, \quad z_2 \sim N(0, \tfrac{2}{3}) \\
X_3 &= \frac{1}{3}X_2^2 + z_3, \quad z_3 \sim N(0, \tfrac{1}{2}).
\end{aligned}
$$

Here we have unequal residual variances. We now check that this model satisfies the conditions in Theorem B.2. Let $f(u) = u^2$ and note that the true ordering is $X_1 \prec X_2 \prec X_3$. Starting from the

Figure 3: Experiments confirming Example 6. For the identifiable setting with $\sigma_3^2 = 1/2$, Algorithm 2 correctly learns the topological ordering. For the non-identifiable setting with $\sigma_3^2 = 1/3$, Algorithm 2 fails to learn the ordering.

first source node $X_1$, we have

$$\sigma_1^2 = 1$$
$$\sigma_2^2 + \mathbb{E}\operatorname{var}(f(X_{\pi(2)})) = 2/3 + \operatorname{var}(X_1^2/2) = 2/3 + 1/2 > 1 = \sigma_1^2$$
$$\sigma_3^2 + \mathbb{E}\operatorname{var}(f(X_{\pi(3)})) = 1/2 + \operatorname{var}(X_2^2/3)$$
$$= 1/2 + \frac{1}{9}\Big(\operatorname{var}(X_1^4)/16 + \operatorname{var}(z_2^2) + \operatorname{var}(X_1^2 z_2)$$
$$+ 2\operatorname{cov}(X_1^4/4, X_1^2 z_2) + 2\operatorname{cov}(z_2^2, X_1^2 z_2)\Big)$$
$$= 1/2 + 8/9 + 8/81 > 1 = \sigma_1^2$$

Then for the second source node $X_2$,

$$\sigma_2^2 = 2/3$$
$$\sigma_3^2 + \mathbb{E}\operatorname{var}(f(X_{\pi(3)}) \mid X_1) = 1/2 + \frac{1}{9}(\operatorname{var}(z_2^2) + \mathbb{E}\operatorname{var}(X_1^2 z_2 \mid X_1))$$
$$= 1/2 + 1/3 - 1/81 > 2/3$$

Thus the condition is satisfied.

If instead we have $\sigma_3^2 = \operatorname{var}(z_3) = 1/3$, the condition would be violated. It is easy to check that nothing changes for $X_1$. For the second source node $X_2$, things are different:

$$\sigma_3^2 + \mathbb{E}\operatorname{var}(f(X_{\pi(3)}) \mid X_1) = 1/3 + 1/3 - 1/81 < 2/3$$

Thus, the order of $X_2$ and $X_3$ would be flipped for this model.

This example is easily confirmed in practice. We let $n$ range from 50 to 1000, and check if the estimated order is correct for the two models ($\sigma_3^2 = 1/2$ and $\sigma_3^2 = 1/3$). We simulated this 50 times and report the averages in Figure 3.

*Proof of Theorem B.2.* We first consider the case where every node in the same layer has a different residual variance, i.e. $X_u, X_v \in L_m \implies \sigma_u^2 \neq \sigma_v^2$. We proceed with induction on the element $j$ of the ordering $\pi$. Let $i = \pi_j$,

When $j = 1$ and $i = \pi_1$, $X_i$ must be a source node and we have for all $k \in \pi_{[2:d]}$,

$$\begin{aligned}
\mathrm{var}(X_i) = \sigma_i^2 <& \sigma_k^2 + \mathbb{E}\,\mathrm{var}(\mathbb{E}(X_k \mid \mathrm{pa}(k))) \\
=& \mathbb{E}\,\mathrm{var}(X_k \mid \mathrm{pa}(k)) + \mathrm{var}(\mathbb{E}(X_k \mid \mathrm{pa}(k))) \\
=& \mathrm{var}(X_k).
\end{aligned}$$

Thus the first node to be identified must be $i = \pi_1$, as desired.

Now suppose the the first $j - 1$ nodes in the ordering $\pi$ are correctly identified. The parent of node $i = \pi_j$ must have been identified in $\pi_{[j-1]}$ or it is a source node. Then we have for all $k \in \{\pi_{j+1}, \ldots, \pi_d\}$,

$$\begin{aligned}
\mathbb{E}\,\mathrm{var}(X_i \mid X_{\pi_{[1:j-1]}}) = \sigma_i^2 <& \sigma_k^2 + \mathbb{E}\,\mathrm{var}(\mathbb{E}(X_k \mid \mathrm{pa}(k)) \mid X_{\pi_{[1:j-1]}}) \\
=& \mathbb{E}\,\mathrm{var}(X_k \mid \mathrm{pa}(k)) + \mathbb{E}\,\mathrm{var}(\mathbb{E}(X_k \mid \mathrm{pa}(k)) \mid X_{\pi_{[1:j-1]}}) \\
=& \mathbb{E}[\mathbb{E}(\mathrm{var}(X_k \mid \mathrm{pa}(k)) \mid X_{\pi_{[1:j-1]}})] + \mathbb{E}[\mathrm{var}(\mathbb{E}(X_k \mid \mathrm{pa}(k)) \mid X_{\pi_{[1:j-1]}})] \\
=& \mathbb{E}\,\mathrm{var}(X_k \mid X_{\pi_{[1:j-1]}})
\end{aligned}$$

Then the $j^{\text{th}}$ node to be identified must be $i = \pi_j$. The induction is completed.

Finally, if $X_u, X_v \in L_m$ are in the same layer and $\sigma_u^2 = \sigma_v^2$, then this procedure may choose either $X_u$ or $X_v$ first. For example, if $X_u$ is chosen first, then $X_v$ will be incorporated into the same layer as $X_u$. Since both these nodes are in the same layer, swapping $X_u$ and $X_v$ in any ordering still produces a valid ordering of the DAG. The proof is complete. $\qquad\square$

## C   Proof of Theorem 4.1

The proof of Theorem 4.1 will be broken down into several steps. First, we derive an upper bound on the error of the plug-in estimator used in Algorithm 2 (Appendix C.1), and then we derive a uniform upper bound (Appendix C.2). Based on this upper bound, we prove Theorem 4.1 via Proposition C.4 (Appendix C.3). Appendix C.4 collects various technical lemmas that are used throughout.

### C.1   A plug-in estimate

Let $(X, Y) \in \mathbb{R}^m \times \mathbb{R}$ be a pair of random variables and $\widehat{f}$ be a data-dependent estimator of the conditional expectation $\mathbb{E}[Y \mid X] := f(X)$. Assume we have split the sample into two groups, which we denote by $(U^{(1)}, V^{(1)}), \ldots, (U^{(n_1)}, V^{(n_1)}) \sim \mathbb{P}(X, Y)$ and $(X^{(1)}, Y^{(1)}), \ldots, (X^{(n_2)}, Y^{(n_2)}) \sim \mathbb{P}(X, Y)$ for clarity. Given these samples, define an estimator of $\sigma_{\mathrm{RV}}^2 := \mathbb{E}\,\mathrm{var}(Y \mid X)$ by

$$\widehat{\sigma}_{\mathrm{RV}}^2 := \frac{1}{n_2}\sum_{i=1}^{n_2}(Y^{(i)})^2 - \frac{1}{n_2}\sum_{i=1}^{n_2}\widehat{f}(X^{(i)})^2. \tag{7}$$

Note here that $\widehat{f}$ depends on $(U^{(i)}, V^{(i)})$, and is independent of the second sample $(X^{(i)}, Y^{(i)})$. We wish to bound the deviation $\mathbb{P}(|\widehat{\sigma}_{\mathrm{RV}}^2 - \sigma_{\mathrm{RV}}^2| \geq t)$.

Define the target $\theta^* = \mathbb{E}f^2(X)$ and its plug-in estimator

$$\theta(g; q) = \int g(x)^2 \, \mathrm{d}q(x).$$

Letting $\mathbb{P}_X$ denote the true marginal distribution with respect to $x$, we have $\theta^* = \theta(f; \mathbb{P}_X)$ and $\widehat{\theta} := \theta(\widehat{f}; \widehat{\mathbb{P}})$, where $\widehat{\mathbb{P}} = n_2^{-1}\sum_i \delta_{X^{(i)}}$ is the empirical distribution. We will also make use of more general targets $\theta(g; \mathbb{P}_X) = \mathbb{E}g^2(X)$ for general functions $g$.

Finally, as a matter of notation, we adopt the following convention: For a random variable $Z$, $\|Z\|_p := (\mathbb{E}_Z|Z|^p)^{1/p}$ is the usual $L^p$-norm of $Z$ as a random variable, and for a (nonrandom) function $f$, $\|f\|_p := (\int |f|^p \, \mathrm{d}\zeta)^{1/p}$, where $\zeta$ is a fixed base measure such as Lebesgue measure. In particular, $\|f(X)\|_p \neq \|f\|_p$. The difference of course lies in which measure integrals are taken with respect to. Moreover, we shall always explicitly specify with respect to which variables probabilities and expectations are taken, e.g. to disambiguate $\mathbb{E}_{\widehat{f}, X} = \mathbb{E}_{\widehat{f}}\mathbb{E}_X$, $\mathbb{E}_{\widehat{f}}$, and $\mathbb{E}_X$.

We first prove the following result:

**Proposition C.1.** *Assume* $\|f(X)\|_\infty, \|\widehat{f}(X)\|_\infty \leq B_\infty$. *Then*

$$\mathbb{P}(|\widehat{\theta} - \theta^*| \geq 2t) \lesssim \frac{\mathbb{E}_{\widehat{f}} \|f(X) - \widehat{f}(X)\|_2^2}{t^2} + \frac{\|f(X)\|_4^4 + \mathbb{E}_{\widehat{f}} \|\widehat{f}(X) - f(X)\|_4}{n_2 t^2}. \tag{8}$$

*Proof.* We have

$$\mathbb{P}_{\widehat{f},X}(|\widehat{\theta} - \theta^*| \geq 2t) \leq \mathbb{P}_{\widehat{f},X}(|\theta^* - \theta(\widehat{f}; \mathbb{P}_X)| \geq t) + \mathbb{P}_{\widehat{f},X}(|\widehat{\theta} - \theta(\widehat{f}; \mathbb{P}_X)| \geq t).$$

The second term is easy to dispense with since

$$\mathbb{P}_{\widehat{f},X}(|\widehat{\theta} - \theta(\widehat{f}; \mathbb{P}_X)| \geq t) \leq \frac{\mathbb{E}_{\widehat{f}} \mathbb{E}_X(\widehat{\theta} - \theta(\widehat{f}; \mathbb{P}_X))^2}{t^2} \leq \frac{\mathbb{E}_{\widehat{f}} \operatorname{var}_X(\widehat{f}^2(X))}{n_2 t^2}. \tag{9}$$

It follows from Lemma C.5 with $p = 4$ that

$$|\mathbb{E}_X \widehat{f}(X)^4 - \mathbb{E}_X f(X)^4| \leq \|\widehat{f}(X) - f(X)\|_4 \sum_{k=0}^{3} \|\widehat{f}(X)\|_4^k \|f(X)\|_4^{3-k}$$

$$\leq 4(2B_\infty)^3 \|\widehat{f}(X) - f(X)\|_4$$

and hence

$$\operatorname{var}_X(\widehat{f}^2(X)) \leq \mathbb{E}_X \widehat{f}^4(X) \lesssim \|f(X)\|_4^4 + \|\widehat{f}(X) - f(X)\|_4. \tag{10}$$

Combined with (9), we finally have

$$\mathbb{P}(|\widehat{\theta} - \theta(\widehat{f}; \mathbb{P}_X)| \geq t) \lesssim \frac{\|f(X)\|_4^4 + \mathbb{E}_{\widehat{f}} \|\widehat{f}(X) - f(X)\|_4}{n_2 t^2}. \tag{11}$$

For the first term, since $f(X), \widehat{f}(X) \in L^\infty$, Lemma C.7 implies

$$\mathbb{E}_{\widehat{f}}(\theta^* - \theta(\widehat{f}; \mathbb{P}_X))^2 = \mathbb{E}_{\widehat{f}} \mathbb{E}_X(f^2(X) - \widehat{f}^2(X))^2$$

$$\lesssim \mathbb{E}_{\widehat{f}} \|f(X) - \widehat{f}(X)\|_2^2,$$

and thus

$$\mathbb{P}_{\widehat{f}}(|\theta^* - \theta(\widehat{f}; \mathbb{P}_X)| \geq t) \leq \frac{\mathbb{E}_{\widehat{f}}(\theta^* - \theta(\widehat{f}; \mathbb{P}_X))^2}{t^2} \lesssim \frac{\mathbb{E}_{\widehat{f}} \|f(X) - \widehat{f}(X)\|_2^2}{t^2}.$$

Therefore

$$\mathbb{P}_{\widehat{f},X}(|\widehat{\theta} - \theta^*| \geq 2t) \leq \mathbb{P}_{\widehat{f},X}(|\theta^* - \theta(\widehat{f}; \mathbb{P}_X)| \geq t) + \mathbb{P}_{\widehat{f},X}(|\widehat{\theta} - \theta(\widehat{f}; \mathbb{P}_X)| \geq t)$$

$$\lesssim \frac{\mathbb{E}_{\widehat{f}} \|f(X) - \widehat{f}(X)\|_2^2}{t^2} + \frac{\|f(X)\|_4^4 + \mathbb{E}_{\widehat{f}} \|\widehat{f}(X) - f(X)\|_4}{n_2 t^2}. \qquad \square$$

Finally, we conclude the following:

**Corollary C.2.** *If* $\|f(X)\|_\infty, \|\widehat{f}(X)\|_\infty \leq B_\infty$, *then*

$$\mathbb{P}(|\widehat{\sigma}_{\mathrm{RV}}^2 - \sigma_{\mathrm{RV}}^2| \geq t) \lesssim \frac{4}{t^2} \left( \mathbb{E}_{\widehat{f}} \|f(X) - \widehat{f}(X)\|_2^2 + \frac{\operatorname{var}(Y) + \|f(X)\|_4^4 + \mathbb{E}_{\widehat{f}} \|\widehat{f}(X) - f(X)\|_4}{n_2} \right).$$

## C.2   A uniform bound

For any $j = 1, \ldots, r$ and $\ell \notin A_j$, define $\sigma_{\ell j}^2 := \mathbb{E} \operatorname{var}(X_\ell \mid A_j)$ and $\widehat{\sigma}_{\ell j}^2$ the corresponding plug-in estimator from (7). By Proposition C.2, we have for $f_{\ell j}(X_{A_j}) := \mathbb{E}[X_\ell \mid X_{A_j}]$,

$$\mathbb{P}(|\widehat{\sigma}_{\ell j}^2 - \sigma_{\ell j}^2| \geq t) \lesssim \frac{4}{t^2} \Big( \mathbb{E}_{\widehat{f}} \|f_{\ell j}(X_{A_j}) - \widehat{f}_{\ell j}(X_{A_j})\|_2^2$$

$$+ \frac{\operatorname{var}(X_\ell) + \|f_{\ell j}(X_{A_j})\|_4^4 + \mathbb{E}_{\widehat{f}} \|\widehat{f}_{\ell j}(X_{A_j}) - f_{\ell j}(X_{A_j})\|_4}{n_2} \Big) \tag{12}$$

Thus we have the following result:

**Proposition C.3.** *Assume for all $j$ and $\ell \notin A_j$:*

1. *$\|f_{\ell j}(X_{A_j})\|_\infty, \|\widehat{f}_{\ell j}(X_{A_j})\|_\infty \leq B_\infty$;*

2. *$\|f_{\ell j}(X_{A_j})\|_4^4 + \operatorname{var}(X_\ell) \leq B_\infty$;*

3. *$\mathbb{E}_{\widehat{f}}\|f_{\ell j}(X_{A_j}) - \widehat{f}_{\ell j}(X_{A_j})\|_2^2 \to 0$.*

*Then*

$$\sup_{\ell,j} \mathbb{P}(|\widehat{\sigma}_{\ell j}^2 - \sigma_{\ell j}^2| > t) \lesssim \frac{4}{t^2}\left(\mathbb{E}_{\widehat{f}}\|f_{\ell j}(X_{A_j}) - \widehat{f}_{\ell j}(X_{A_j})\|_2^2 + \frac{1}{n_2}\right). \tag{13}$$

For example, under Conditions 1-3, we have $B_\infty := \sup_{\ell,j} 2\|f_{\ell j}(X_{A_j})\|_\infty^4 + \zeta_0$.

## C.3 Proof of Theorem 4.1

Recall $\sigma_{\ell j}^2 = \mathbb{E}\operatorname{var}(X_\ell \mid A_j)$ and $\widehat{\sigma}_{\ell j}^2$ is the plug-in estimator defined by (7). Let $\xi > 0$ be such that

$$\sup_{\ell,j} \mathbb{P}(|\widehat{\sigma}_{\ell j}^2 - \sigma_{\ell j}^2| > t) \leq \frac{\xi^2}{t^2}. \tag{14}$$

For example, Proposition C.3 implies $\xi^2 \asymp \delta^2 + n_2^{-1}$. Recall also $\Delta := \inf_j \Delta_j$, where $\Delta_j > 0$ is the smallest number such that $\mathbb{E}\operatorname{var}(X_\ell \mid A_j) > \sigma^2 + \Delta_j$ for all $\ell \notin A_j$.

Theorem 4.1 follows immediately from Proposition C.4 below, combined with Proposition C.3 to bound $\xi$ by $\delta$.

**Proposition C.4.** *Define $\xi > 0$ as in (14). Then for any threshold $\xi\sqrt{d} < \eta < \Delta/2$, we have*

$$\mathbb{P}(\widehat{L} = L(\mathsf{G})) \geq 1 - \frac{\xi^2}{\eta^2} rd.$$

*Proof.* Define $\mathcal{E}_{j-1} := \{\widehat{L}_1 = L_1, \ldots, \widehat{L}_{j-1} = L_{j-1}\}$. It follows that

$$\mathbb{P}(\widehat{L} = L(\mathsf{G})) = \mathbb{P}(\widehat{L}_1 = L_1, \ldots, \widehat{L}_r = L_r) = \prod_{j=1}^r \mathbb{P}(\widehat{L}_j = L_j \mid \mathcal{E}_{j-1}).$$

Clearly, if $\widehat{L}_1 = L_1, \ldots, \widehat{L}_r = L_r$ then $\widehat{r} = r$. By definition, we have $\sigma_{\ell j}^2 > \sigma^2 + \Delta$, i.e. $\Delta$ is the smallest "gap" between any source in a subgraph $\mathsf{G}[V - A_{j-1}]$ and the rest of the nodes.

Now let $\widehat{\sigma}^2 := \min_\ell \widehat{\sigma}_{\ell 0}^2$ and consider $L_1$:

$$\mathbb{P}(\widehat{L}_1 = L_1) = \mathbb{P}(|\widehat{\sigma}_{\ell 0}^2 - \widehat{\sigma}^2| \leq \eta \,\forall \ell \in A_1, |\widehat{\sigma}_{\ell 0}^2 - \widehat{\sigma}^2| > \eta \,\forall \ell \notin A_1)$$

Now for any $k, \ell \in L_j$,

$$|\widehat{\sigma}_{\ell j}^2 - \widehat{\sigma}_{kj}^2| \leq |\widehat{\sigma}_{\ell j}^2 - \sigma^2| + |\widehat{\sigma}_{kj}^2 - \sigma^2|,$$

and for any $\ell \notin L_j$ and $k \in L_j$,

$$|\widehat{\sigma}_{\ell j}^2 - \widehat{\sigma}_{kj}^2| > \Delta_j - |\sigma_{\ell j}^2 - \widehat{\sigma}_{\ell j}^2| - |\widehat{\sigma}_{kj}^2 - \sigma_{kj}^2|.$$

Thus, with probability $1 - d\xi^2/t^2$, we have

$$|\widehat{\sigma}_{\ell j}^2 - \widehat{\sigma}_{kj}^2| \leq 2t \quad \text{if } k, \ell \in L_j, \text{ and}$$
$$|\widehat{\sigma}_{\ell j}^2 - \widehat{\sigma}_{kj}^2| > \Delta_j - 2t \quad \text{if } \ell \notin L_j \text{ and } k \in L_j.$$

Now, as long as $t < \Delta/4$, we have $\Delta_j - 2t > 2t$, which implies that $\eta := 2t < \Delta/2$.

Finally, we have

$$\mathbb{P}(\widehat{L}_1 \neq L_1) \leq \frac{\xi^2}{\eta^2} d \implies \mathbb{P}(\widehat{L}_1 = L_1) \geq 1 - \frac{\xi^2}{\eta^2} d.$$

Recall that $d_j := |L_j|$. Then by a similar argument

$$\mathbb{P}(\widehat{L}_2 = L_2 \mid \widehat{L}_1 = L_1) \geq 1 - \frac{\xi^2}{\eta^2}(d - d_1).$$

Recalling $\mathcal{E}_{j-1} := \{\widehat{L}_1 = L_1, \ldots, \widehat{L}_{j-1} = L_{j-1}\}$, we have just proved that $\mathbb{P}(\widehat{L}_2 = L_2 \mid \mathcal{E}_1) \geq 1 - (d - d_1)(\xi^2/\eta^2)$. A similar argument proves that $\mathbb{P}(\widehat{L}_j = L_j \mid \mathcal{E}_{j-1}) \geq 1 - (\xi^2/\eta^2)(d - d_{j-1})$. Since $\eta > \xi\sqrt{d}$, the inequality $\prod_j (1 - x_j) \geq 1 - \sum_j x_j$ implies

$$\begin{aligned}
\mathbb{P}(\widehat{L} = L(\mathsf{G})) &= \prod_{j=1}^{r} \mathbb{P}(\widehat{L}_j = L_j \mid \mathcal{E}_{j-1}) \\
&= \prod_{j=1}^{r} \left(1 - \frac{\xi^2}{\eta^2}(d - d_{j-1})\right) \\
&\geq 1 - \sum_{j=1}^{r} \frac{\xi^2}{\eta^2}(d - d_{j-1}) \\
&\geq 1 - \frac{\xi^2}{\eta^2}rd
\end{aligned}$$

as desired. $\qquad\square$

## C.4 Technical lemmas

**Lemma C.5.**

$$|\mathbb{E}X^p - \mathbb{E}Y^p| \leq \|X - Y\|_p \sum_{k=0}^{p-1} \|X\|_p^k \|Y\|_p^{p-1-k}$$

*Proof.* Write $\mathbb{E}X^p - \mathbb{E}Y^p$ as a telescoping sum:

$$\begin{aligned}
|\mathbb{E}X^p - \mathbb{E}Y^p| = |\|X\|_p^p - \|Y\|_p^p| &= |\|X\|_p - \|Y\|_p| \cdot \sum_{k=0}^{p-1} \|X\|_p^k \|Y\|_p^{p-k-1} \\
&\leq \|X - Y\|_p \cdot \sum_{k=0}^{p-1} \|X\|_p^k \|Y\|_p^{p-k-1}. \qquad\square
\end{aligned}$$

**Lemma C.6.** *Fix $p > 2$ and $\delta > 0$ and suppose $\|f\|_{p+\delta}, \|g\|_{p+\delta} \leq B_{p+\delta}$. Then*

$$\|f - g\|_p \leq C_{p,\delta} \cdot \|f - g\|_2^{\gamma_{p,\delta}}, \quad C_{p,\delta} = (2B_{p+\delta})^{\frac{(p-2)(p+\delta)}{p(p+\delta-2)}}, \quad \gamma_{p,\delta} = \frac{2\delta}{p(p+\delta-2)}.$$

*The exponent $\gamma_{p,\delta}$ satisfies $\gamma_{p,\delta} \leq 2/p < 1$ and $\gamma_{p,\delta} \to 2/p$ as $\delta \to \infty$, and the constant $C_{p,\delta} \to 1 - \frac{2}{p}$ as $\delta \to \infty$. Thus, if $f - g \in L^\infty$, then*

$$\|f - g\|_p \lesssim \|f - g\|_2^{2/p}.$$

*Proof.* Use log-convexity of $L^p$-norms with $2 = q < p < r = p + \delta$. $\qquad\square$

**Lemma C.7.** *Assume $\|f\|_\infty \leq B_\infty < \infty$ and $g \in L^4$. Then*

$$\mathbb{E}_X(f(X)^2 - g(X)^2)^2 \leq \|f(X) - g(X))\|_4^4 + \\
4B_\infty \|f(X) - g(X)\|_3^3 + 4B_\infty^2 \|f(X) - g(X)\|_2^2. \tag{15}$$

*If additionally $g(X) \in L^\infty$, then*

$$\mathbb{E}_X(f(X)^2 - g(X)^2)^2 \lesssim \|f(X) - g(X)\|_2^2. \tag{16}$$

*Proof.* Note that

$$(f(X)^2 - g(X)^2)^2 = (g(X) - f(X))^4 + 4f(X)(g(X) - f(X))^3 + 4f(X)^2(g(X) - f(X))^2$$
$$\leq (g(X) - f(X))^4 + 4|f(X)||g(X) - f(X)|^3 + 4f(X)^2(g(X) - f(X))^2.$$

Thus

$$\mathbb{E}(f(X)^2 - g(X)^2)^2 \leq \mathbb{E}(g(X) - f(X))^4 +$$
$$4\mathbb{E}[|f(X)||g(X) - f(X)|^3] + 4\mathbb{E}[f(X)^2(g(X) - f(X))^2]$$
$$\leq \mathbb{E}(g(X) - f(X))^4 +$$
$$4B_\infty \mathbb{E}|g(X) - f(X)|^3 + 4B_\infty^2 \mathbb{E}(g(X) - f(X))^2.$$

This proves the first inequality. The second follows from taking $\delta \to \infty$ in Lemma C.6 and using $\|f - g\|_p^p \lesssim \|f - g\|_2^2$ for $p = 3, 4$. $\qquad\square$

# D   Comparison to CAM algorithm

In this section, we justify the claim in Example 5 that *there exist infinitely many nonlinear functions g for which the CAM algorithm returns an incorrect graph under the model* (4). To show this, we first construct a *linear* model on which CAM returns an incorrect ordering. Since CAM focuses on nonlinear models, we then show that this extends to any sufficiently small nonlinear perturbation of this model.

The linear model is

$$\begin{cases} X_1 \sim \mathcal{N}(0, 1) \\ X_2 = X_1 + z_2 \quad z_2 \sim \mathcal{N}(0, 1) \\ X_3 = X_1 + X_2 + z_3 \quad z_3 \sim \mathcal{N}(0, 1). \end{cases} \tag{17}$$

The graph is

$$X_1 \to X_2$$
$$\searrow \downarrow$$
$$X_3$$

which corresponds to the adjacency matrix

$$\begin{pmatrix} 0 & 1 & 1 \\ 0 & 0 & 1 \\ 0 & 0 & 0 \end{pmatrix}.$$

The *IncEdge* step of the CAM algorithm (§5.2, [7]) is based on the following score function:

$$\sum_{j=1}^d \log\left(\mathbb{E}\,\mathrm{var}(X_j \mid \mathrm{pa}(j))\right).$$

The algorithm starts with the empty DAG (i.e. $\mathrm{pa}(j) = \emptyset$ for all $j$) and proceeds by greedily adding edges that decrease this score the most in each step. For example, in the first step, CAM searches for the pair $(X_i, X_j)$ that maximizes $\log \mathrm{var}(X_j) - \log \mathbb{E}\,\mathrm{var}(X_j \mid X_i)$, and adds the edge $X_i \to X_j$ to the estimated DAG. The second proceeds similarly until the estimated order is determined. Thus, it suffices to study the log-differences $\omega(j, i, S) := \log \mathrm{var}(X_j \mid X_S) - \log \mathbb{E}\,\mathrm{var}(X_j \mid X_{S \cup i})$.

The following are straightforward to compute for the model (17):

$$\mathrm{var}(X_1) = 1 \quad \mathbb{E}\,\mathrm{var}(X_2|X_1) = 1 \quad \mathbb{E}\,\mathrm{var}(X_1|X_2) = \frac{1}{2}$$
$$\mathrm{var}(X_2) = 2 \quad \mathbb{E}\,\mathrm{var}(X_3|X_1) = 2 \quad \mathbb{E}\,\mathrm{var}(X_1|X_3) = \frac{3}{2}$$
$$\mathrm{var}(X_3) = 6 \quad \mathbb{E}\,\mathrm{var}(X_3|X_2) = \frac{1}{3} \quad \mathbb{E}\,\mathrm{var}(X_2|X_3) = \frac{1}{2}$$

Figure 4: The CAM algorithm does not recover the correct ordering under different nonlinear functions and models. $h(x)$ refers to model (19), $g(x)$ refers to model (4) respectively.

Then

$$\log\left(\frac{\text{var}(X_2)}{\mathbb{E}\,\text{var}(X_2|X_1)}\right) = \log\left(\frac{2}{1}\right) = \log 2 \quad \log\left(\frac{\text{var}(X_1)}{\mathbb{E}\,\text{var}(X_1|X_2)}\right) = \log\left(\frac{1}{1/2}\right) = \log 2$$

$$\log\left(\frac{\text{var}(X_3)}{\mathbb{E}\,\text{var}(X_3|X_1)}\right) = \log\left(\frac{6}{2}\right) = \log 3 \quad \log\left(\frac{\text{var}(X_1)}{\mathbb{E}\,\text{var}(X_1|X_3)}\right) = \log\left(\frac{1}{1/3}\right) = \log 3$$

$$\log\left(\frac{\text{var}(X_3)}{\mathbb{E}\,\text{var}(X_3|X_2)}\right) = \log\left(\frac{6}{3/2}\right) = \log 4 \quad \log\left(\frac{\text{var}(X_2)}{\mathbb{E}\,\text{var}(X_2|X_3)}\right) = \log\left(\frac{2}{1/2}\right) = \log 4$$

Now, if $X_3 \to X_2$ is chosen first, then the order is incorrect and we are done. Thus suppose CAM instead chooses $X_2 \to X_3$, then in the next step it would update the score for $X_1 \to X_3$ to be

$$\log\left(\frac{\mathbb{E}\,\text{var}(X_3|X_2)}{\mathbb{E}\,\text{var}(X_3|X_1, X_2)}\right) = \log\left(\frac{3/2}{1}\right) = \log\frac{3}{2} < \log\left(\frac{\mathbb{E}\,\text{var}(X_1|X_2)}{\mathbb{E}\,\text{var}(X_1|X_3, X_2)}\right) = \log 3$$

Therefore, for the next edge, CAM would choose $X_3 \to X_1$, which also leads to the wrong order. Thus regardless of which edge is selected first, CAM will return the wrong order.

Thus, when CAM is applied to data generated from (17), it is guaranteed to return an incorrect ordering. Although the model (17) is identifiable, it does not satisfy the identifiability condition for CAM (Lemma 1, [7]), namely that the structural equation model is a nonlinear additive model. Thus, we need to extend this example to an identifiable, nonlinear additive model.

Since this depends only on the scores $\omega(j, i, S)$, it suffices to construct a *nonlinear* model with similar scores. For this, we consider a simple nonlinear extension of (17): Let $g$ be an arbitrary bounded, nonlinear function, and define $g_\delta(u) := u + \delta g(u)$. The nonlinear model is given by

$$\begin{cases} X_1 \sim \mathcal{N}(0, 1) \\ X_2 = g_\delta(X_1) + z_2 & z_2 \sim \mathcal{N}(0, 1) \\ X_3 = g_\delta(X_1) + g_\delta(X_2) + z_3 & z_3 \sim \mathcal{N}(0, 1). \end{cases} \tag{18}$$

This model satisfies both our identifiability condition (Condition 2) and the identifiability condition for CAM (Lemma 1, [7]).

We claim that for sufficiently small $\delta$, the CAM algorithm will return the wrong ordering (see Proposition D.1 below for a formal statement). It follows that the scores $\omega(j, i, S; \delta)$ corresponding to the model (18) can be made arbitrarily close to $\omega(j, i, S) = \omega(j, i, S; 0)$, which implies that CAM will return the wrong ordering for sufficiently small $\delta > 0$.

In Figure 4, we illustrate this empirically. In addition to the model (18), we also simulated from the following model, which shows that this phenomenon is not peculiar to the construction above:

$$\begin{cases} X_1 \sim \mathcal{N}(0,1) \\ X_2 = X_1^2 + z_2 \quad z_2 \sim \mathcal{N}(0,1) \\ X_3 = 4X_1^2 + h(X_2) + z_3 \quad z_3 \sim \mathcal{N}(0,1). \end{cases} \tag{19}$$

In all eight examples, NPVAR perfectly recovers the ordering while CAM is guaranteed to return an inconsistent order for sufficiently large $n$ (i.e. once the scores are consistently estimated).

**Proposition D.1.** *Let $\mathbb{E}_\delta$ and $\mathrm{var}_\delta$ be taken with respect to model (18). Then for all $i,j \in \{1,2,3\}$, as $\delta \to 0$,*

$$|\mathrm{var}_\delta(X_i) - \mathrm{var}_0(X_i)| = o(1),$$
$$|\mathbb{E}_\delta \mathrm{var}_\delta(X_i|X_j) - \mathbb{E}_0 \mathrm{var}_0(X_i|X_j)| = o(1).$$

*and $|\mathbb{E}_\delta \mathrm{var}_\delta(X_3|X_1,X_2) - \mathbb{E}_0 \mathrm{var}_0(X_3|X_1,X_2)| = o(1)$.*

*Proof sketch of Proposition D.1.* The proof is consequence of the fact that the differences are continuous functions of $\delta$. We sketch the proof for $i = 2$; the remaining cases are similar.

We have

$$\mathrm{var}_\delta(X_2) = \mathrm{var}_\delta(g(X_1)) + \mathrm{var}_\delta(\epsilon)$$
$$\mathrm{var}_0(X_2) = \mathrm{var}_0(X_1) + \mathrm{var}_0(\epsilon)$$

Let $\varphi(t)$ be the standard normal density. We only need to analyze and compare $\mathrm{var}_0(X_1) - \mathrm{var}_\delta(g(X_1)) = \mathrm{var}_0(X_1) - \mathrm{var}_0(g(X_1))$ in two parts:

$$\int (X_1^2 - g(X_1)^2)\varphi(X_1)dX_1$$
$$\left(\int X_1\varphi(X_1)dX_1\right)^2 - \left(\int g(X_1)\varphi(X_1)dX_1\right)^2.$$

Since $|X_1 - g(X_1)| \le \delta$,

$$|\int (X_1^2 - g(X_1)^2)\varphi(X_1)dX_1| \le \delta \int |X_1 + g(X_1)|\varphi(X_1)dX_1 \le \delta \int 2|X_1|\varphi(X_1)dX_1 + \delta^2 = C\delta + \delta^2$$

$$|\int g(X_1)\varphi(X_1)dX_1| = |\int (X_1 - g(X_1))\varphi(X_1)dX_1| \le \delta \int \varphi(X_1)dX_1 = \delta$$

$$|\int (X_1 + g(X_1)\varphi(X_1))dX_1| = |\mathbb{E}g(X_1)| \le \delta.$$

Thus

$$|\left(\int X_1\varphi(X_1)dX_1\right)^2 - \left(\int g(X_1)\varphi(X_1)dX_1\right)^2| \le \delta^2,$$

so that

$$|\mathrm{var}_\delta(X_2) - \mathrm{var}_0(X_2)| = o(1)$$

as claimed. □

# E   Experiment details

In this appendix we outline the details of our experiments, as well as additional simulations.

## E.1   Experiment settings

For graphs, we used

- *Markov chain (MC)*. Graph where there is one edge from $X_{i-1}$ to $X_i$ for all nodes $i = 2, \ldots, d$.

- *Erdős Rényi (ER)*. Random graphs whose edges are added independently with specified expected number of edges.
- *Scale-free networks (SF)*. Networks simulated according to the Barabasi-Albert model.

For models, we refer to the nonlinear functions in SEM. We specify the nonlinear functions in $X_j = f_j(X_{\text{pa(j)}}) + z_j$ for all $j = 1, 2, \ldots, d$, where $z_j \overset{\text{iid}}{\sim} \mathcal{N}(0, \sigma^2)$ with variance $\sigma^2 \in \{0.2, 0.5, 0.8\}$

- *Additive sine model (SIN)*: $f_j(X_{\text{pa}(j)}) = \sum_{k \in \text{pa}(j)} f_{jk}(X_k)$ where $f_{jk}(X_k) = \sin(X_k)$.
- *Additive Gaussian process (AGP)*: $f_j(X_{\text{pa}(j)}) = \sum_{k \in \text{pa}(j)} f_{jk}(X_k)$ where $f_{jk}$ is a draw from Gaussian process with RBF kernel with length-scale one.
- *Non-additive Gaussian process (NGP)*: $f_j$ is a draw from Gaussian process with RBF kernel with length-scale one.
- *Generalized Linear Model (GLM)*: This is a special case with non-additive model and non-additive noise. We specify the model $\mathbb{P}(X_j = 1) = f_j(X_{\text{pa}(j)})$ by a parameter $p \in \{0.1, 0.3\}$. Given $p$, if $j$ is a source node, $\mathbb{P}(X_j = 1) = p$. For the Markov chain model we define:

$$f_j(X_{\text{pa}(j)}) = \begin{cases} p & X_k = 1 \\ 1 - p & X_k = 0 \end{cases}$$

  or vice versa.

We generated graphs from each of the above models with $\{d, 4d\}$ edges each. These are denoted by the shorthand XX-YYY-$k$, where XX denotes the graph type, YYY denotes the model, and $k$ indicates the graphs have $kd$ edges on average. For example, ER-SIN-1 indicates an ER graph with $d$ (expected) edges under the additive sine model. SF-NGP-4 indicates an SF graph with $4d$ (expected) edges under a non-additive Gaussian process. Note that there is no difference between additive or non-additive GP for Markov chains, so the results for MC-NGP-$k$ are omitted from the Figures in Appendix E.5.

Based on these models, we generated random datasets with $n$ samples. For each simulation run, we generated $n \in \{100, 200, 500, 750, 1000\}$ samples for graphs with $d \in \{5, 10, 20, 40, 50, 60, 70\}$ nodes.

### E.2 Implementation and baselines

Code implementing NPVAR can be found at https://github.com/MingGao97/NPVAR. We implemented NPVAR (Algorithm 2) with generalized additive models (GAMs) as the nonparametric estimator for $\widehat{f}_{\ell j}$. We used the gam function in the R package mgcv with P-splines bs='ps' and the default smoothing parameter sp=0.6. In all our implementations, including our own for Algorithm 2, we used default parameters in order to avoid skewing the results in favour of any particular algorithm as a result of hyperparameter tuning.

We compared our method with following approaches as baselines:

- Regression with subsequent independence test (RESIT) identifies and disregards a sink node at each step via independence testing [43]. The implementation is available at http://people.tuebingen.mpg.de/jpeters/onlineCodeANM.zip. Uses HSIC test for independence testing with alpha=0.05 and gam for nonparametric regression.
- Causal additive models (CAM) estimates the topological order by greedy search over edges after a preliminary neighborhood selection [7]. The implementation is available at https://cran.r-project.org/src/contrib/Archive/CAM/. By default, CAM applies an extra pre-processing step called preliminary neighborhood search (PNS). Uses gam to compute the scores and for pruning, and mboost for preliminary neighborhood search. The R implementation of CAM does not use the default parameters for gam or mboost, and instead optimizes these parameters at runtime.
- NOTEARS uses an algebraic characterization of DAGs for score-based structure learning of nonparametric models via partial derivatives [61, 62]. The implementation is available at

https://github.com/xunzheng/notears. We used neural networks for the nonlineari-
ties with a single hidden layer of 10 neurons. The training parameters are `lambda1=0.01`,
`lambda2=0.01` and the threshold for adjacency matrix is `w_threshold=0.3`.

- Greedy equivalence search with generalized scores (GSGES) uses gen-
eralized scores for greedy search without assuming model class [20].
The implementation is available at https://github.com/Biwei-Huang/
Generalized-Score-Functions-for-Causal-Discovery/. We used cross-
validation parameters `parameters.kfold = 10` and `parameters.lambda = 0.01`.

- Equal variance (EqVar) algorithm identifies source node by minimizing conditional variance
in linear SEM [8]. The implementation is available at https://github.com/WY-Chen/
EqVarDAG. The original EqVar algorithm estimates the error variances in a linear SEM
via the covariance matrix $\Sigma = \mathbb{E}XX^T$, and then uses linear regression (e.g. best subset
selection) to learn the structure of the DAG. We adapted this algorithm to the nonlinear
setting in the obivous way by using GAMs (instead of subset selection) for variable selection.
The use of the covariance matrix to estimate the order remains the same.

- PC algorithm and greedy equivalence search (GES) are standard baselines for structure learn-
ing [10, 52]. The implementation is available from R package `pcalg`. For PC algorithm, use
correlation matrix as sufficient statistic. Independence test is implemented by `gaussCItest`
with significance level `alpha=0.01`. For GES, set the score to be `GaussL0penObsScore`
with `lambda` as $\log n/2$, which corresponds to the BIC score.

The experiments were conducted on an Intel E5-2680v4 2.4GHz CPU with 64 GB memory.

### E.3  Metrics

We evaluated the performance of each algorithm with the following two metrics:

- $\mathbb{P}$(correct order): The percentage of runs in which the algorithm gives a correct topological
ordering, over $N$ runs. This metric is only sensible for algorithms that first estimate an
ordering or return an adjacency matrix which does not contain undirected edges, including
RESIT, CAM, EqVar and NOTEARS.

- Structural Hamming distance (SHD): A standard benchmark in the structure learning lit-
erature that counts the total number of edge additions, deletions, and reversals needed to
convert the estimated graph into the true graph.

Since there may be multiple topological orderings of a DAG, in our evaluations of order recovery,
we check whether or not the order returned is any of the possible valid orderings. For PC, GES, and
GSGES, they all return a CPDAG that may contain undirected edges, in which case we evaluate
them favourably by assuming correct orientation for undirected edges whenever possible. Since
CAM, RESIT, EqVar and NPVAR each first estimate a topological ordering then estimate a DAG.
To estimate a DAG from an ordering, we apply the same pruning step to each algorithm for a fair
comparison, which is adapted from [7]. Specifically, given an estimated ordering, we run a `gam`
regression for each node on its ancestors, then determine the parents of the node by the $p$-values with
significance level `cutoff=0.001` for estimating the DAG.

### E.4  Timing

For completeness, runtime comparisons are reported in Tables 1 and 2. Algorithms based on linear
models such as EqVar, PC, and GES are by far the fastest. These algorithms are also the most highly
optimized. The slowest algorithms are GSGES and RESIT. Timing comparisons against CAM are
are difficult to interpret since by default, CAM first performs preliminary neighbourhood search,
which can easily be applied to any of the other algorithms tested. The dramatic difference with
and without this pre-processing step by comparing Tables 1 and 2: For $d = 40$, with this extra
step CAM takes just over 90 seconds, whereas without it, on the same data, it takes over 8.5 hours.
For comparison, NPVAR takes around two minutes (i.e. without pre-processing or neighbourhood
search).

| Algorithm | $d$ | $n$ | Runtime (s) |
|---|---|---|---|
| EqVar | 20 | 1000 | $0.0017 \pm 0.0003$ |
| PC | 20 | 1000 | $0.056 \pm 0.016$ |
| GES | 20 | 1000 | $0.060 \pm 0.034$ |
| NPVAR | 20 | 1000 | $10.76 \pm 0.23$ |
| NOTEARS | 20 | 1000 | $31.46 \pm 8.79$ |
| CAM (w/ PNS) | 20 | 1000 | $40.56 \pm 1.29$ |
| CAM (w/o PNS) | 20 | 1000 | $559.01 \pm 9.49$ |
| RESIT | 20 | 1000 | $652.15 \pm 7.26$ |
| GSGES | 20 | 1000 | $3216.00 \pm 95.0$ |

Table 1: Runtime comparisons for $d = 20$. Timing for CAM is presented with and without preliminary neighbourhood selection (PNS), which is a pre-processing step that can be applied to any algorithm. In our experiments, only CAM used PNS.

| Algorithm | $d$ | $n$ | Runtime (s) |
|---|---|---|---|
| EqVar | 40 | 1000 | $0.0043 \pm 0.0003$ |
| GES | 40 | 1000 | $0.12 \pm 0.0052$ |
| PC | 40 | 1000 | $0.019 \pm 0.030$ |
| NOTEARS | 40 | 1000 | $76.05 \pm 19.16$ |
| CAM (w/ PNS) | 40 | 1000 | $95.59 \pm 6.33$ |
| NPVAR | 40 | 1000 | $118.33 \pm 2.25$ |
| CAM (w/o PNS) | 40 | 1000 | $31644.56 \pm 1329.31$ |

Table 2: Runtime comparisons for $d = 40$. Timing for CAM is presented with and without preliminary neighbourhood selection (PNS), which is a pre-processing step that can be applied to any algorithm. In our experiments, only CAM used PNS.

### E.5  Additional experiments

Here we collect the results of our additional experiments. Since the settings MC-AGP-$k$ and MC-NGP-$k$ are equivalent (i.e. since there is only parent for each node), the plots for MC-NGP-$k$ are omitted. Some algorithms might be skipped due to high computational cost or numerical issue.

- Figure 5: SHD vs. $n$ with $d = 5$ fixed, across all graphs and models tested.
- Figure 6: SHD vs. $n$ with $d = 10$ fixed, across all graphs and models tested.
- Figure 7: SHD vs. $n$ with $d = 20$ fixed, across all graphs and models tested.
- Figure 8: SHD vs. $n$ with $d = 40$ fixed, across all graphs and models tested with GSGES and RESIT skipped (due to high computational cost).
- Figure 9: SHD vs. $n$ with $d = 50$ fixed, across all graphs and models tested with GSGES, RESIT and NOTEARS skipped (due to high computational cost).
- Figure 10: SHD vs. $n$ with $d = 60$ fixed, across all graphs and models tested with GSGES, RESIT and NOTEARS skipped (due to high computational cost).
- Figure 11: SHD vs. $n$ with $d = 70$ fixed, across all graphs and models tested with GSGES, RESIT and NOTEARS skipped (due to high computational cost).
- Figure 12: SHD vs. $n$ with $d$ ranging from 5 to 70 on GLM with CAM and RESIT skipped (due to numerical issues).
- Figure 13: SHD vs. $d$ with $n = 500$ fixed, across all graphs and models tested.
- Figure 14: SHD vs. $d$ with $n = 1000$ fixed, across all graphs and models tested.
- Figure 15: Ordering recovery vs. $n$ with $d = 5$ fixed, across all graphs and models tested.
- Figure 16: Ordering recovery vs. $n$ with $d = 10$ fixed, across all graphs and models tested.

Figure 5: SHD vs $n$ for fixed $d = 5$.

Figure 6: SHD vs $n$ for fixed $d = 10$.

Figure 7: SHD vs $n$ for fixed $d = 20$.

Figure 8: SHD vs $n$ for fixed $d = 40$.

Figure 9: SHD vs $n$ for fixed $d = 50$.

Figure 10: SHD vs $n$ for fixed $d = 60$.

Figure 11: SHD vs $n$ for fixed $d = 70$.

Figure 12: SHD vs $n$ for different $d$ ranging from 5 to 70 on GLM

Figure 13: SHD vs $d$ for fixed $n = 500$.

Figure 14: SHD vs $d$ for fixed $n = 1000$.

Figure 15: Ordering recovery vs $n$ for fixed $d = 5$.

Figure 16: Ordering recovery vs $n$ for fixed $d = 10$.

Figure 17: Ordering recovery vs $n$ for fixed $d = 20$.

Figure 18: Ordering recovery vs $n$ for fixed $d = 40$.

Figure 19: Ordering recovery vs $n$ for fixed $d = 50$.

Figure 20: Ordering recovery vs $n$ for fixed $d = 60$.

Figure 21: Ordering recovery vs $n$ for fixed $d = 70$.