[Reviews · NeurIPS 2020]

Review 1

Summary and Contributions: Update after author rebuttal phase: I stand by my initial evaluation of the paper. I am comfortable with its acceptance at NeurIPS 20. ---------------- This paper considers the problem of learning nonparametric causal DAGs from data. The authors propose an algorithm that works in the nonparametric setting without assumptions on linearity, additivity, independent noise, and faithfulness, and establish computational and sample complexity results for this algorithm under an assumption on residual variances. They also compare the proposed algorithm to prior work via a simulation study

Strengths: - The theoretical claims of the paper are sound and well justified. The empirical evaluation is extensive and well documented. - The problem of learning causal DAGs is very relevant to the NeurIPS community. - the results are novel to the best of my knowledge.

Weaknesses: The entire paper relies on the observation in Theorem 3.1 -- which is a simple generalization of the equal variance property of linear models. This claim is reasonably straightforward to establish, and not particularly surprising. It is this fact that allows one to consider nonparametric DAGs as opposed to linear or additive models. After this is established, the rest of the paper is quite a standard combination of machinery from the DAG and nonparametric statistics literature. Given this, I think the contribution of this work is somewhat incremental.

Correctness: The empirical and theoretical results are correct.

Clarity: The paper is well written. There are hyperlinks that point between the supplementary material and the main document that should probably be deactivated.

Relation to Prior Work: Relation to prior work is sufficiently well addressed.

Reproducibility: Yes

Additional Feedback:


Review 2

Summary and Contributions: The paper proposes a method for learning Bayes nets with continuous variables, for the nonparametric case.

Strengths: The work is novel and relevant. The theoretical results seem sound. The experimental results are encouraging.

Weaknesses: Sparsity was not assumed and thus, the sample complexity is exponential with respect to number of nodes d. (Please see my additional feedback below.)

Correctness: The main claims seem correct.

Clarity: The paper is clear.

Relation to Prior Work: Relation to prior work is properly discussed.

Reproducibility: Yes

Additional Feedback: Sparsity was not assumed and thus, the sample complexity is exponential with respect to number of nodes d. Please discuss how the current proofs can be modified to accomodate for sparsity, and what would be the sample complexity in this case. Minor comment: Some of the Arxiv references are actually papers in conference proceedings. === AFTER REBUTTAL I am satisfied with the authors response regarding several ways to improve the bounds with assumptions such as sparsity. Given the above, I keep my initial evaluation of 7.


Review 3

Summary and Contributions: The paper gives an algorithm that runs in time polynomial in the number of variables d and that learns the data generating DAG with high probability from about O(d^d) samples. The basic idea is to sort the variables based on their estimated residual variances. Empirical results are favorable in comparison to NOTEARS and some other benchmark methods.

Strengths: + Good theoretical results. + Appears to also have practical significance. + Clearly written, with a good number of examples.

Weaknesses: - The large sample size needed for theoretical guarantees makes the result uninformative regarding practical sample sizes. - The theoretical results rely on a condition on residual variances that may not hold in typical practical instances.

Correctness: I could not spot any errors.

Clarity: The clarity of the presentation is satisfactory.

Relation to Prior Work: Relevant prior work is adequately discussed, as far as I can tell.

Reproducibility: Yes

Additional Feedback: ADDENDUM: Thank you for the rebuttal. You may want to emphasize why polynomial time complexity should be of interest when the sample complexity is exponential (reading the data takes exponential time already). I agree that you have managed to remove a lot of typical assumptions in an interesting manner (even if extending previously presented ideas) -- with the cost of adding a somewhat questionable new assumption about the residual variances.


Review 4

Summary and Contributions: This paper presents a polynomial-time algorithm for learning directed acyclic graphs (DAGs). This paper extends work for learning graphs from data with equal variances to data where the conditional variance of X_i does not depend on i.

Strengths: - The examples given in 3.1 are very helpful. - The experiment section is strong.

Weaknesses: - The title says "causal" but the main results are about DAGs. As the authors state, DAGs may be interpreted causally under additional assumptions. Is "causal" really appropriate in the title? - I do not understand what is meant by "simultaneous statistical and computational guarantees" as stated in the introduction. Is it just that computational complexity in 3.2 and sample complexity in section 4 are both given? - The algorithm makes just a simple change to the that of citation [5].

Correctness: Yes.

Clarity: Mostly yes. Some parts are confusing or potentially misleading, as described above.

Relation to Prior Work: This is discussed, but it is not clear what are the assumptions needed by previous work on equal variances? Does that work assume lineariry, additivity, independent noise, or faithfulness?

Reproducibility: Yes

Additional Feedback: UPDATE: I have changed my review to 6 based on discussion period.

[Author Response · NeurIPS 2020]

We thank all three reviewers for their time and feedback. Below we have done our best to respond to the major concerns.

A common concern was the surprising simplicity of the method. It is crucial to note that prior to our work, there was
*no provably poly-time algorithm with sample complexity guarantees* for this problem in a nonparametric setting. It is
difficult to overstate this point: Our analysis is completely model-free and comes with explicit guarantees. Existing
analyses fail on nonparametric models (see **(A1)**), and existing nonparametric methods do not come with poly-time
guarantees. This is the main contribution of our work, to resolve this open problem and prove that such an algorithm
exists (L31-33; see also L1-4; L24-29; L36-37; L71-73; L181-182; L198-L202). Despite the similarity to existing work,
our algorithm is *not* the same, and indeed several subtle but crucial changes were made to eliminate reliance on linearity,
additivity, and independence of noise (see **(A1)**).

**(A1) [R1, R4] Incremental.** The analysis from existing work fails on nonparametric models, and our analysis is
*completely* different—we analyze a *different* algorithm with a *different* technical approach. We must contrast Alg 1
and Alg 2: Alg 1 is a direct translation of existing algorithms (refs [5,11,12,32]) for which existing proofs *fail* when
applied to nonparametric models, whereas Alg 2 contains crucial changes to adapt to the nonparametric setting, namely
the use of the layer decomposition and sample splitting. We can show with an explicit counterexample that existing
proofs would not generalize to Alg 1: If Alg 1 is applied to the null DAG model (no edges), then one needs to bound
the estimation error for all $d2^{d-1}$ possible residual variances (this example is not unique or pathological; any DAG with
more than one sort will have similar issues). This is subtle: Essentially, Alg 1 randomly chooses $O(d^2)$ parameters
based on the data (note the estimated $\widehat{A}_j$ in Alg 2). This is precisely the reason for modifying Alg 1 into Alg 2: By
learning the DAG layer-by-layer, this combinatorial explosion is avoided. By contrast, existing work crucially relies on
linearity to write the residual variances in terms of the covariance matrix $\Sigma$ (L167-171) in order to bound all $d2^{d-1}$
choices uniformly. For nonparametric models, there is no such representation via $\Sigma$, and each residual variance must be
estimated separately. Regrettably this discussion was missing and we will add it to the camera ready version.

Furthermore, our analysis is nontrivial in several aspects: Our main results do not depend on any specific regression
estimator, which uses several interesting tools (e.g. log-convexity and interpolation of $L^p$ norms; see Appendix C.4).
This makes our results more practical. The proof of Theorem 4.1 is also completely different from related work on
equal variances, and informs the modifications made in Alg 2. See **(A2)** for a discussion of the novelty of Theorem 3.1.

**(A2) [R1, R4] Relation to prior work.** Prior work on equal variances crucially relies on linearity and independent,
additive noise; see L102-105. Linearity is not crucial for identifiability, but is leveraged extensively to obtain statistical
guarantees; please see **(A1)** for more details. One of our contributions is to show that these assumptions can be
*completely* removed, without qualification: Our results apply to arbitrary nonlinear models with correlated, non-additive
noise. For example, although the proof of Theorem 3.1 is straightforward, it is not quite a "simple generalization" of
existing work: Our proof is completely different, and proves something much stronger using only the Markov property
of BNs. We emphasize that existing results *completely miss this*, arguably because they rely on independence and
additivity of noise in a *crucial* way (L96-101), though it is not needed. Faithfulness is not required by previous work on
equal variances, but is commonly assumed in other work on BNs. We are happy to add this discussion to the paper.

**(A3) [R2, R3] Sparsity and sample complexity.** As pointed out at L250-257, there are several ways to improve the
sample complexity. The most direct approach is to use a more sophisticated estimator of $\mathbb{E}\,\mathrm{var}(X_\ell \,|\, X_A)$, for which
faster (root-$n$) rates are available (ref [9]; see also L308-311). Another approach, as suggested by R2, is to assume
some kind of sparsity: By using adaptive estimators such as RODEO [21] or GRID [13], the sample complexity
will depend only on the sparsity of $f_{\ell j}(X_{A_j})$, i.e. $d^* = \max_j \max_{\ell \notin A_j} |\{k \in A_j : \partial_k f_{\ell j} \neq 0\}|$ ($\partial_k$ is the $k$th
partial derivative). Here is another way that does not require adaptive estimation: Suppose $|L_j| \leq w$ and define
$r^* := \sup\{|i - j| : e = (e_1, e_2) \in E, e_1 \in L_i, e_2 \in L_j\}$. Then $\delta^2 \asymp n^{-2/(2+wr^*)}$, and the resulting sample
complexity depends on $wr^*$ instead of $d$. For a Markov chain with $w = r^* = 1$ this leads to a substantial improvement.

**(A4) [R3] Assumptions.** Certainly our main assumption may not hold in some practical situations. We have gone to
great lengths to address this: (a) Our results hold in *far greater generality than existing work*, not requiring linearity,
additivity, independent noise, or faithfulness (L3-4, L24-25, L181-182); (b) Our algorithm provably recovers models
that state-of-the-art algorithms fail to recover (Sec 3.3, Ex 5); (c) The main assumption can be substantially weakened
to unequal variances (L151-157, App B.1); (d) We have additional experiments on misspecified models in App B.1.

**(A5) [R4] Causality / title.** Here we are following convention in the literature, which refers to this problem as "causal
discovery", "causal DAG learning", etc. We are happy to add more details on this point, e.g. by including a discussion
of causal assumptions such as minimality and noting that under our assumptions there is a unique causal DAG.

**(A6) [R4] Guarantees.** A notable drawback of existing work is the failure to accomplish *both* nonasymptotic statistical
and algorithmic guarantees in nonparametric settings. We have devoted an entire section (Sec 3.3) to highlight this
point (see also L198-202). Most estimators for nonparametric DAG models come with one or the other: Finite-sample
statistical guarantees that lack efficiency guarantees, or poly-time guarantees without explicit sample complexities. We
are happy to add additional comparisons to emphasize this point.

[Meta-Review · NeurIPS 2020]

After discussions, the majority of reviewers think that the contributions are enough to learn causal graphs, even though some points should be made clearer in the final version (including the fact that causality may come with a stronger meaning than it is used and details on the assumptions). The response from authors has been important, but has not been appreciate by all reviewers - I suggest a better approach in the future. Moreover, I do not think that the optional field to send comments to the AC is meant to be used to write yet another rebuttal. These points could have jeopardized this submission.